# The Framework for Assessing Changes To Sea-level (FACTS) v1.0: A platform for characterizing parametric and structural uncertainty in future global, relative, and extreme sea-level change

Robert E. Kopp[1,2], Gregory G. Garner[1,2,*], Tim H. J. Hermans[3,4], Shantenu Jha[2,5,6], Praveen Kumar[1,2], Alexander Reedy[1,2,5], Aimée B.A. Slangen[3], Matteo Turilli[5,6], Tamsin L. Edwards[7], Jonathan M. Gregory[8,9], George Koubbe[5], Anders Levermann[10,11], Andre Merzky[5], Sophie Nowicki[12], Matthew D. Palmer[8,13], and Chris Smith[14,15]

[1]Department of Earth and Planetary Sciences, Rutgers University, Piscataway, NJ, USA
[2]Rutgers Climate and Energy Institute, Rutgers University, New Brunswick, NJ, USA
[3]NIOZ Royal Netherlands Institute for Sea Research, Department of Estuarine & Delta Systems, Yerseke, The Netherlands
[4]Utrecht University, Institute for Marine and Atmospheric research Utrecht (IMAU), Utrecht, The Netherlands
[5]Department of Electrical and Computer Engineering, Rutgers University, Piscataway, NJ, USA
[6]Computational Science Initiative, Brookhaven National Laboratory, Upton, NY, USA
[7]Department of Geography, King's College London, London, UK
[8]Met Office Hadley Centre, Exeter, UK
[9]National Centre for Atmospheric Science, University of Reading, Reading, UK
[10]Potsdam Institute for Climate Impact Research, Potsdam, Germany
[11]Physics Institute, Potsdam University, Potsdam, Germany
[12]University at Buffalo, Buffalo, NY, USA
[13]University of Bristol, School of Earth Sciences, Bristol, UK
[14]Priestley International Centre for Climate, University of Leeds, UK
[15]International Institute for Applied Systems Analysis (IIASA), Laxenburg, Austria
[*]Current affiliation: Gro Intelligence, New York, NY, USA

**Correspondence:** Robert E. Kopp (robert.kopp@rutgers.edu)

**Abstract.** Future sea-level rise projections are characterized by both quantifiable uncertainty and unquantifiable, structural uncertainty. Thorough scientific assessment of sea-level rise projections requires analysis of both dimensions of uncertainty. Probabilistic sea-level rise projections evaluate the quantifiable dimension of uncertainty; comparison of alternative probabilistic methods provides an indication of structural uncertainty. Here we describe the Framework for Assessing Changes To Sea-level (FACTS), a modular platform for characterizing different probability distributions for the drivers of sea-level change and their consequences for global-mean, regional, and extreme sea-level change. We demonstrate its application by generating seven alternative probability distributions under multiple emissions scenarios for both future global-mean sea-level change and future relative and extreme sea-level change at New York City. These distributions, closely aligned with those presented in the Intergovernmental Panel on Climate Change Sixth Assessment Report, emphasize the role of the Antarctic and Greenland ice sheets as drivers of structural uncertainty in sea-level change projections.

## 1 Introduction

Quantitative projections of future sea-level change have been of interest to both scientists and decision-makers since at least the 1980s (Garner et al., 2018; Hall et al., 2019; Horton et al., 2018; Kopp et al., 2019). To our knowledge, the first peer-reviewed scientific article projecting 21st century global-mean sea-level rise appeared in *Science* in 1982 (Gornitz et al., 1982). The US Army Corps of Engineers and the Dutch Ministerie van Verkeer en Waterstaat first employed planning-oriented sea-level scenarios just four years later, in 1986 (US Army Corps of Engineers, 1986; National Research Council, 1987; van der Kley, 1987). Thus, sea-level projections have always been one of the more practically relevant parts of scientific assessments of climate change, including all six of the Intergovernmental Panel on Climate Change (IPCC) Working Group 1 reports (Kopp et al., 2023).

At the same time, scientific projections of sea-level rise have also long acknowledged the presence of factors – particularly associated with Antarctic ice-sheet instability – that limit the ability to generate quantitative sea-level projections (e.g., Mercer, 1978; Gornitz et al., 1982). These limits give rise to what is sometimes called ambiguity or deep uncertainty – uncertainty that cannot be represented by singular probability distributions, due to limited amount, reliability, and unanimity of information (ambiguity as defined by Ellsberg, 1961) or, similarly, to ignorance or disagreement among analysts (subtypes of deep uncertainty as defined by Lempert et al., 2003). The question of how to integrate such ambiguity into the assessment and communications of sea-level projections has long challenged the authors of scientific assessments (Oppenheimer et al., 2019b; Kopp et al., 2023).

Until about fifteen years ago, comprehensive, localized projections of relative sea-level (RSL) change were uncommon (e.g., Katsman et al., 2008; National Research Council, 2012).[1] Many users simply augmented global-mean sea level (GMSL) projections with estimates of vertical land motion (VLM) to project local RSL change. The IPCC Fourth Assessment Report (AR4) considered only deviations from GMSL driven by ocean dynamic sea-level change as represented in coupled atmosphere-ocean general circulation models (Meehl et al., 2007). Other researchers focused on the contemporary gravitational, rotational, and deformational (GRD) RSL changes caused by redistribution of mass within the cryosphere and hydrosphere (e.g., by melting land ice) (e.g., Mitrovica et al., 2009). These two threads began to come together in the literature leading up to the IPCC Fifth Assessment Report (AR5) (e.g., Kopp et al., 2010; Slangen et al., 2012). AR5 was the first IPCC report to consider both sterodynamic sea-level change and contemporary GRD, along with the effects of glacial isostatic adjustment (GIA), in its RSL projections (Church et al., 2013a).

The AR5 projections and numerous subsequent studies taking on the challenge of producing comprehensive, localized RSL projections (see section 9.6.3.1 of Fox-Kemper et al., 2021a, for an overview) are generally referred to as 'probabilistic' projections, in that, under different emissions scenarios, they estimate probability distributions for the change in each of the driving factors of GMSL and RSL change and their total. Producing such projections requires combining different lines of

---

[1]See Box 1 for sea-level terminology.

information: global climate models (GCMs) can simulate sterodynamic sea-level change, but do not in general include coupled glaciers, ice sheets, or anthropogenic changes in land water storage. They also require using relatively simple representations of sea-level drivers; models of the complexity of GCMs do not lend themselves to the Monte Carlo sampling used to estimate sea-level distributions in probabilistic sea-level projections. Examples of open-source probabilistic sea-level projection frameworks include the ProjectSL/LocalizeSL framework (Kopp and Rasmussen, 2021), developed by Kopp et al. (2014, 2017), and BRICK (Wong et al., 2017). Additional studies present probabilistic RSL projection methodologies without associated open-source software releases (e.g., Slangen et al., 2014; Grinsted et al., 2015; Jackson and Jevrejeva, 2016; Jevrejeva et al., 2019; Le Cozannet et al., 2019; Palmer et al., 2020).

Probabilistic sea-level projection frameworks are limited in that they assume that future changes under a single emissions scenario can be represented by a single probability distribution. By definition, this assumption is not true for processes characterized by ambiguity (Kopp et al., 2023; Hinkel et al., 2019). While some studies (e.g., Kopp et al., 2017; Jevrejeva et al., 2019) have worked around this problem to explore structural uncertainties by substituting different modeling approaches for different sea-level components, probabilistic projection frameworks have not generally been engineered to facilitate such explorations.

This paper describes the Framework for Assessing Changes To Sea-level (FACTS), a scalable, modular, open-source framework for global-mean, local, and extreme sea-level projection that is designed to support the characterization of ambiguity in sea-level projections. FACTS is built using modern computational practices and in the spirit of open science (e.g., Wilkinson et al., 2016). It is designed so users can easily explore deep uncertainty by investigating the implications for GMSL, RSL, and extreme sea level (ESL) of different choices for different processes. Its modularity allows components to be represented by either simple or complex models. Because it is built upon the RADICAL-Cybertools computing stack (Merzky et al., 2021), different modules can in principle be dispatched for execution on resources appropriate to their computational complexity.

FACTS is, specifically, a tool for sea-level *assessment*. It is not intended as a substitute for detailed, process-based analyses of individual sea-level contributions (for example, GCM studies of ocean dynamics, or ice-sheet modeling studies) or of integrated projections made with high-complexity Earth system models that are moving toward including coupled ice sheets (e.g., Muntjewerf et al., 2021; Smith et al., 2021). Such studies provide the scientific bases underlying FACTS modules. Rather, FACTS is intended to support scientists – like those participating in the IPCC and in numerous national and subnational assessment processes – who seek to develop projections that are internally consistent, represent the richness of approaches present in the scientific literature, and assess multiple types of uncertainty. Such assessment outputs, rather than individual projections in the primary scientific literature, are generally the primary way in which climate risk practitioners interact with estimates of future sea-level change (Kopp et al., 2023).

Development versions of FACTS modules underlie the GMSL and RSL projections of the IPCC Sixth Assessment Report (AR6) (Fox-Kemper et al., 2021a; Slangen et al., 2023) and the 2022 US Government sea-level rise Technical Report (Sweet et al., 2022). For these implementations, several key steps were run offline, and modules were invoked outside the execution and data management framework provided by the FACTS Manager. FACTS 1.0 allows replication of the AR6 approach entirely within FACTS, starting from specification of emissions scenarios and ending with the production of multiple, alternative probability distributions for GMSL, RSL and ESL.

---

**Box 1. Key sea-level terminology**

This paper employs terminology for sea level based on Gregory et al. (2019).

**Contemporary GRD:** *GRD* due to ongoing changes in the mass of water stored on land in the cryosphere and hydrosphere.

**Extreme sea level (ESL):** The occurrence or the level of an exceptionally high or low local sea-surface height. FACTS models high ESLs, which can be caused, for example, by storm surges or exceptionally high tides.

**Geoid:** A surface on which the geopotential has a uniform value, chosen so that the volume enclosed between the geoid and the sea floor is equal to the time-mean volume of sea water in the ocean (including the liquid-water equivalent of floating ice).

**Glacial isostatic adjustment (GIA):** *GRD* due to ongoing changes in the solid Earth caused by past changes in land ice.

**Global-mean sea-level (GMSL) rise:** The increase in the volume of the ocean divided by the ocean surface area.

**Global-mean thermosteric sea-level rise:** The part of *GMSL rise* which is due to thermal expansion.

**Gravitational, rotational and deformational (GRD) effects:** Changes in Earth gravity and Earth rotation (which alter the shape of the *geoid* and thus sea-surface height), as well as viscoelastic solid-Earth deformation (which causes *VLM* and, by altering the shape of the ocean basin, affects sea-surface height).

**Inverse barometer (IB) effect:** The time-dependent hydrostatic depression of the sea surface by atmospheric pressure variations. In most GCMs, atmospheric pressure variations are not communicated to the ocean. In such GCMs, the change in the *IB effect* must be added to the change in simulated sea-surface height in order to produce a quantity comparable to observed *RSL change*.

**Ocean dynamic sea-level change:** The change in the mean local height of the sea surface above the geoid, excluding the change in the *IB effect*.

**Relative sea-level (RSL) change:** The change in local mean sea level relative to the local solid surface, i.e., the sea floor.

**Sterodynamic sea-level change:** *RSL change* due to changes in ocean density and circulation. Sterodynamic sea-level change is equal to the sum of *global-mean thermosteric sea-level change* and *ocean dynamic sea-level change*. Note that the quantity projected by the `tlm/sterodynamics` module of FACTS is the sum of *sterodynamic sea-level change* and the climatological change in the *IB effect*.

**Vertical land movement (VLM):** The change in the height of the sea floor or the land surface.

---

## 2 Model Description

### 2.1 Overview

FACTS consists of the FACTS Manager, which oversees the execution of FACTS experiments, and an extendable suite of modules, which provide the scientific and analytical core that allow FACTS to simulate the different process contributing to GMSL, RSL and ESL change. Modules represent independent processes (e.g., sterodynamic sea-level change or VLM) and can

be run in parallel on high-performance computing (HPC) resources. Modules can also be run in sequence when their outputs depend upon inputs from other modules (e.g., the modules that compute total RSL change and ESL distribution shifts).

A FACTS experiment consists of a series of Experiment Steps (Figure 1). Typical Experiment Steps include: (1) a climate Experiment Step, which translates an inputted emissions scenario into projections of global-mean surface air temperature (GSAT) and ocean heat content change; (2) a sea-level components Experiment Step, which simulates the different physical processes driving sea-level change; (3) an integration Experiment Step, which adds up the different components into projections of total GMSL and RSL change; and (4) an ESL Experiment Step, which uses tide gauge data and RSL projections to project the change in extreme sea-level occurrences over time.

Each Experiment Step runs one or more modules in parallel. Exchange of information between modules happens in between Experiment Steps. This exchange is mediated by the file system, so Experiment Steps can be bypassed simply by providing appropriate input files (e.g, stored GSAT and ocean heat content trajectories) to the subsequent Experiment Step. Though the existing usage of FACTS contains only one sea-level component Experiment Step, and therefore treats the output of each module as independent conditional upon their common dependence on the climate simulated in the climate Experiment Step, the FACTS Manager allows Experiment Steps to be subdivided and thus could support between-module coupling.

The core concept of Workflow provides FACTS with the flexibility required to explore structural uncertainty. A Workflow consists of a set of sea-level component modules that are added together in the integration Experiment Step to produce a probabilistic estimate of their combined contribution to sea-level change. Workflows can be overlapping: for example, two Workflows might use the same module for simulating sterodynamic sea-level change, but use different modules for simulating ice sheet change. Modules run in the sea-level components Experiment Step are tagged as belonging to one or more Workflows; those Workflows are then aggregated at the integration Experiment Step. This structure allows a single sea-level components Experiment Step to include multiple modules representing alternative methods to simulate the same sea-level component and avoids redundant execution of modules employed in multiple Workflows.

In practice, for a specific set of climate inputs (e.g., emissions scenario-forced GSAT projections), a single Workflow produces a single (climate input-conditional) probabilistic projection of sea-level change. Multiple Workflows can be compared to examine the structural uncertainty of GMSL, RSL, and ESL change to the choice of component methods (i.e., the ambiguity of projections) and combined (for example, in a p-box, as discussed in section 4.1) to produce summary outputs that capture ambiguity (Kopp et al., 2023).

## 2.2 FACTS Manager and RADICAL-Cybertools

Though most of the FACTS modules implemented to date can be run on a desktop computer, and all can run on small-scale HPC clusters, FACTS is designed to allow modules of a broad range of computational demands, including those requiring supercomputer resources. This objective is achieved by using the RADICAL-Cybertools software stack in the FACTS Manager.

RADICAL-Cybertools are software systems designed to support the execution, across computing scales, of applications comprised of multiple tasks. A task can be any executable or Python function; tasks can have short ($\mathcal{O}$(seconds)) or long ($\mathcal{O}$(hours to days)) duration and can run on single or multiple cores, nodes, and threads, either locally or remotely.

RADICAL Ensemble-Toolkit (hereafter, EnTK) (Balasubramanian et al., 2016, 2018) is the top-level system of the middle-ware stack used to implement FACTS. EnTK is an ensemble execution system, implemented as a Python library, that offers components to encode and execute ensemble applications on HPC systems. EnTK uses RADICAL-Pilot (RP) (Merzky et al., 2021) to decouple the description of ensemble applications from their execution, separating three concerns: (i) specification of tasks and resource requirements; (ii) resource selection and acquisition; and (iii) management of task execution. EnTK sits between the user and the HPC system(s), abstracting resource and execution management complexities from the user.

EnTK exposes an API with three user-facing constructs: Pipeline, Stage, and Task. Those constructs allow the user to encode an ensemble application in terms of concurrency and sequentiality of tasks. Each Pipeline is a sequence of Stages, and each Stage is a set of Tasks. Consistent with their formal definition, EnTK executes the members of a set concurrently and the members of a sequence sequentially. For example, all the Stages of each Pipeline execute sequentially, and all the Tasks of each Stage execute concurrently. In this way, EnTK describes an ensemble application in terms of the concurrency and sequentiality of tasks, without requiring the explicit specification of tasks' data or control dependencies.

In the context of the FACTS Manager, each Experiment Step contains a set of Pipelines that are run concurrently. Each Pipeline is associated with one FACTS module, and each module runs a series of sequential, single-Task Stages described in its configuration file. Most typically, these Stages consist of: (1) a pre-processing Stage with a Task that prepares associated data; (2) a fitting Stage with a Task that calibrates the module based on the data prepared by the pre-processing Stage; (3) a projection Stage; and (4) a post-processing Stage. In the existing sea-level component modules, the projection Stage generates the projection of GMSL contributions, while the post-processing Stage generates the projection of RSL contributions. For example, in a module computing Greenland ice sheet contributions, the projection Stage might project the Greenland contribution to GMSL, while the post-processing Stage might incorporate the contemporary GRD effects that modulate the Greenland contribution to RSL change at specific sites. Note that alternative specifications are possible; e.g., the totaling module runs in a single Stage.

## 2.3 Modules

FACTS 1.0 includes a library of different modules (Table 1) that both illustrate functionality and allow simulation of projection work flows analogous to those employed in the IPCC AR6 (Fox-Kemper et al., 2021a). Each of the included modules is described below. Configuration options such as the number of samples to run, the time points at which calculations are reported, and the reference period used for output can be globally specified but are implemented on a module-by-module basis.

### 2.3.1 Climate module

Climate simulation is provided by the `fair/temperature` module. This module wraps around the FaIR v1.6.4 climate model emulator (Smith et al., 2018; Millar et al., 2017), using the AR6 calibrated and constrained parameter set (Smith, 2021). Taking an emissions scenario as an input, this module samples uncertainty in key climate model parameters (e.g., equilibrium climate sensitivity and transient climate response) and generates probability distributions of GSAT and ocean heat content (using the two-layer temperature function of Geoffroy et al., 2013). The climate simulation Experiment Step can also be bypassed by providing to the modules run in the sea-level components Experiment Step an output file containing these probability

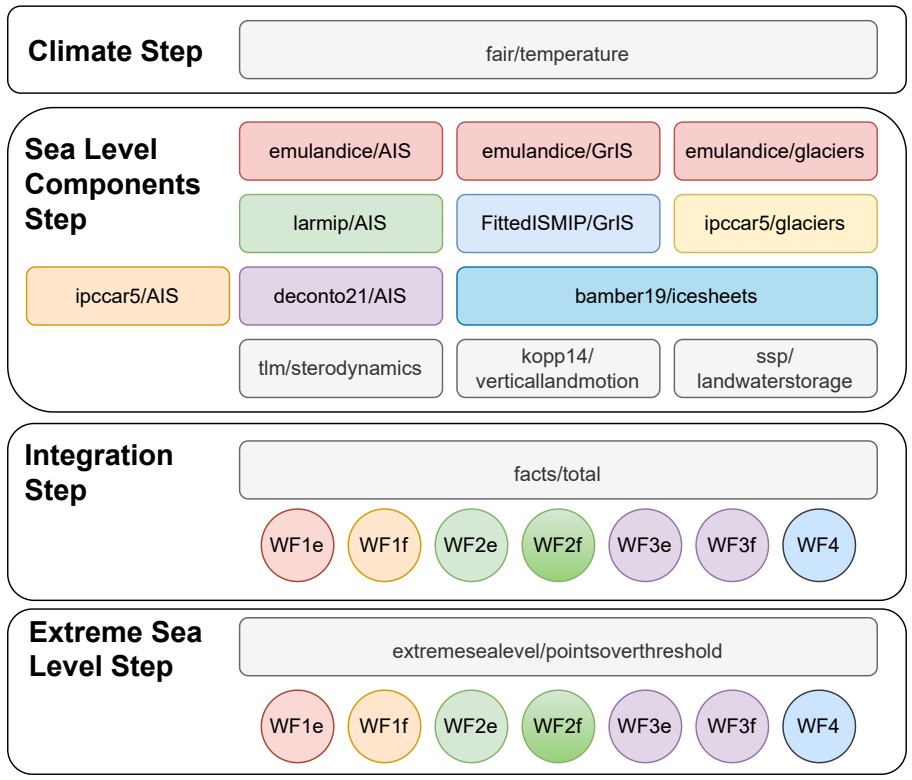

**Figure 1.** Schematic illustration of the FACTS Experiment described in this manuscript. Large boxes represent the four Experiment Steps. Smaller boxes represent different modules run in each Experiment Step. Circles represent Workflows (WF) generated by combining sets of sea-level component modules in the integration Experiment Step (and carried forward to the extreme sea-level Experiment Step). Grey modules are applied to/included in all Workflows, while colored modules are included in some but not all Workflows in different combinations. See Table 2 for details of the modules making up each Workflow.

distributions. For application in the AR6, for example, the climate simulation was run offline and passed as an input to modules depending on these inputs.

### 2.3.2 Sea-level component modules

The bulk of the modules distributed with FACTS simulate physical processes that contribute to GMSL and/or RSL change.
155 Consistent with IPCC AR6, existing sea-level component modules output quantities that are relative to the 19-year average of GMSL and/or RSL centered in the year 2005.

### 2.3.2.1 Generic module

The simplest module in FACTS 1.0 is the generic direct sampling module (`facts/directsample`), which simply translates an ensemble of time series samples specified in a text file into FACTS.

### 2.3.2.2 IPCC AR6 offline land-ice modules

For the implementation of FACTS used to develop the IPCC AR6 sea-level projections, several of the modules used to simulate ice sheet and glacier contributions (`ipccar6/ismipemuicesheets`, `ipccar6/gmipemuglaciers`, `ipccar6/larmipAIS`, `deconto21/AIS`, `bamber19/icesheets`) were based upon variants of `facts/directsample`, with the sample inputs being generated through offline simulation.

In the case of `ipccar6/ismipemuicesheets` and `ipccar6/gmipemuglaciers`, climate output generated offline using FaIR by the AR6 Working Group 1 Chapter 7 authors was run offline through the emulandice emulator of Edwards et al. (2021), the output of which was then transferred to the FACTS modules as static data. Similarly, in the case of `ipccar6/larmipAIS`, the Chapter 7 climate output was run through the LARMIP-2 emulator of Levermann et al. (2020), then transferred to the FACTS module. (Details of both the emulandice and LARMIP-2 emulators are described below.) For replicability reasons, the original AR6 direct-sample version of the emulandice ISMIP6 and LARMIP modules (`ipccar6/ismipemuicesheets` and `ipccar6/larmipAIS`, respectively) are retained in FACTS 1.0, though their use is deprecated.

### 2.3.2.3 Greenland and Antarctic Ice sheet modules

In FACTS 1.0, the `larmip` and `emulandice` modules bring the formerly offline-coupled emulandice and LARMIP-2 emulators into FACTS. These modules are both driven by sampled projections of GSAT.

The `emulandice` modules are structured as wrappers around separately developed, R-language Gaussian process emulators for ISMIP6 ice sheet simulations and GlacierMIP glacier simulations. They demonstrate the ability of FACTS to incorporate independently developed models (Nowicki et al., 2016, 2020; Hock et al., 2019; Marzeion et al., 2020; Edwards et al., 2021). The ISMIP6 (Ice Sheet Model Intercomparison Project for CMIP6: Nowicki et al., 2016, 2020) project generated around 600 simulations from 2015-2100 from 27 modelling groups under very high (RCP8.5/SSP5-8.5) and low (RCP2.6/SSP1-2.6) emissions scenarios, systematically varying a small number of ice-sheet model parameters driving the response. These simulations were used in constructing Gaussian Process emulators of the Greenland and Antarctic (West, East and Peninsula) contributions to sea level as a function of GSAT and of these parameters (Edwards et al., 2021). Note that `emulandice` emulates sea-level contributions in each year independently: the outputs are samples drawn from independent distributions for each year. This means it does not include temporal autocorrelation in uncertainty and therefore does not emulate the rates of change between years, although they can be approximated by smoothing the annual percentiles with a temporal filter (temporal correlation emerges from the underlying simulations). Because the ISMIP6 experiments end in 2100, and Gaussian process emulation should not be used for significant extrapolation (being non-parametric), the `emulandice` modules cannot generate projections beyond 2100.

For the Greenland ice sheet, `FittedISMIP/GrIS` provides a parametric emulator for 21 models participating in the ISMIP6 exercise. The parametric emulator is based on fitting each model's projected sea-level contributions under different scenarios as a cubic function of GSAT and quadratic function of time. Details are provided by Fox-Kemper et al. (2021b). In contrast to `emulandice`, `FittedISMIP/GrIS` can be used to estimate rates of change.

The `larmip` module is an adaptation of separately developed code (Levermann et al., 2020), modified to achieve substantial speed improvements. Within the Linear Antarctic Response Model Intercomparison Project (LARMIP-2), 16 state-of-the-art ice-sheet models performed experiments in which they applied a constant additional basal ice shelf melt forcing of 8 m/yr underneath each of five distinct regions of the Antarctic coast for 200 years. The time derivative of the ice loss response from these experiments yielded a linear response function for each of the regions in each of the models. To apply these linear response functions to generate new projections, GSAT projections are scaled and time-delayed in according with the response of the CMIP6 climate models' subsurface oceanic warming to surface warming. This subsurface warming signal is then scaled with the observed sensitivities of basal melting to warming outside of the Antarctic ice shelf cavities. The resulting basal melt forcing is convolved with the linear response function to project the dynamic response of the Antarctic ice sheet.

Because the LARMIP-2 experiment examined only the dynamic response of the Antarctic ice sheet, projecting the full Antarctic response requires incorporating a separate term representing surface-mass balance changes. This is done within the `larmip` module using the same approach as applied by the IPCC Fifth Assessment Report (Church et al., 2013b) and in the `ipccar5/icesheets` modules, described below. Whereas in AR6, LARMIP-2 projections (including surface mass balance) are extrapolated beyond 2100 assuming a fixed rate of ice-sheet mass loss after 2100, here we allow the rate of loss to evolve following the linear response function formulation.

The `ipccar5/icesheets` module implements the Greenland and Antarctic ice sheet projection methods used in the IPCC Fifth Assessment Report (Church et al., 2013b). Greenland surface mass balance is projected using a cubic polynomial of GSAT (Fettweis et al., 2013). The polynomial is multiplied with a log-normally distributed factor representing methodological uncertainty. Another multiplier varying randomly between 1 and 1.15 is added to account for positive elevation feedback. For Antarctic surface mass balance, accumulation is projected to increase by $5.1 \pm 1.5\%$ per degree Celsius warming in Antarctica, with a $1.1 \pm 0.2$ ratio of warming in Antarctica to GSAT increase. The uncertainties in both of these numbers are assumed to be normally distributed, and a negative rate term that scales with accumulation is added to account for the feedback between enhanced accumulation and dynamic ice discharge. The ice dynamic contributions of Greenland and Antarctica are parameterized by quadratic functions of time, starting at either the lower or upper end of the uncertainty range of observed rates of ice loss over 2005-2010 and reaching respectively the minimum or maximum contributions of the ice sheets in 2100 that the Fifth Assessment Report assessed based on the available literature at that time. Samples are drawn assuming a uniform probability density in between these extreme quadratic functions (Church et al., 2013b).

FACTS 1.0 also includes direct sampling modules used to incorporate ice-sheet projections that include, either by structured expert judgement (`bamber19/icesheets`) (Bamber et al., 2019) or physical modeling (`deconto21/AIS`) (DeConto et al., 2021), processes such as Marine Ice Cliff Instability that are not included in most ice-sheet models but that might have the potential to substantially accelerate the ice sheet contribution to sea level. Bamber et al. (2019) used formal structured expert

judgement with calibrated expert responses to probabilistically evaluate Antarctic and Greenland mass loss through 2300 under
$2°C$ and $5°C$ GSAT stabilization scenarios. In the IPCC AR6, these two GSAT scenarios were mapped to SSP1-2.6 and SSP5-8.5 projections. DeConto et al. (2021) projected future Antarctic ice sheet changes under the Representative Concentration Pathway (RCP) scenarios using a model that incorporates hydrofracturing of ice shelves and the gravitational instability of marine ice cliffs without the protection of a buttressing ice shelf. In the IPCC AR6, the RCP scenario projections were employed in the context of the corresponding SSP projections (e.g., RCP2.6 projections from DeConto et al. (2021) applied to SSP1-2.6).

All the existing ice-sheet modules include in their post-processing Stage a regional scaling based on GRD fingerprints for West Antarctica, East Antarctica, and Greenland (e.g., Mitrovica et al., 2001; Gomez et al., 2010; Mitrovica et al., 2011). The fingerprints include both gravitational and rotational effects on sea-surface height, as well as deformational effects on sea-floor height. They are implemented as static fingerprints that do not change over time; as in Kopp et al. (2014), mass change is assumed to be uniform across the respective regions. The fingerprints were pre-computed (outside the FACTS framework) by solving the sea-level equation with a pseudo-spectral approach up to spherical harmonic degree and order 512 (equivalent to a spatial resolution of about $0.4°$). They assume a radially symmetric, elastic and compressible Earth model based on the Preliminary Reference Earth Model (Dziewonski and Anderson, 1981).

#### 2.3.2.4    Glacier modules

The `emulandice/glacier` module, like the `emulandice/AIS` and `emulandice/GrIS` module, is based on Gaussian process emulation of a multimodel intercomparison exercise, specifically the GlacierMIP2 ensemble (Marzeion et al., 2020), and is driven by inputted GSAT trajectories. The GlacierMIP2 project generated nearly 300 simulations of 2015-2100 glacier loss from 11 modelling groups under four RCP scenarios. These simulations were used in constructing Gaussian Process emulators of the 19 glacier region contributions to sea level as a function of GSAT (Edwards et al., 2021). Because the GlacierMIP experiments end in 2100 (as for the ice sheets), the emulandice modules cannot generate projections beyond 2100.

The `ipccar5/glaciers` module is based on the the glacier projection approach used in the IPCC Fifth Assessment Report (Church et al., 2013b), which models the global-mean sea-level change due to the melt of glaciers as $f \times I(t)^p$, where $I(t)$ is the time-integral of GSAT at time $t$, and $f$ and $p$ are parameters estimated from simulations of a set of four glacier models (Giesen and Oerlemans, 2013; Marzeion et al., 2012; Radić et al., 2014; Slangen and Van De Wal, 2011). The glacier models are equally weighted and systematic uncertainty in the glacier projections is accounted for by Monte Carlo sampling, assuming a normal distribution with a time-dependent model-specific standard deviation. For the glacier projections of the IPCC AR6, these parameters were also derived from the simulations of GlacierMIP and GlacierMIP2 and added as calibration options to the `ipccar5/glaciers` module (Hock et al., 2019; Marzeion et al., 2020; Fox-Kemper et al., 2021a). (In this manuscript, as in IPCC AR6, we focus on the GlacierMIP2 calibration of this module, denoted as GMIP2 in tables). As the IPCC AR5 model itself does not disaggregate the glacier contribution into separate regions, this disaggregation is based upon the time-varying proportion of the contributions of different glaciers in the median projection of Kopp et al. (2014).

As described in Kopp et al. (2014), the `kopp14/glaciers` module projects the contribution of 17 different glaciers and ice cap regions for different RCPs by employing a multivariate t-distribution of ice mass change estimated from the model simulations of (Marzeion et al., 2012) for different source regions.

As with the ice sheet modules, the glacier modules scale their output in the post-processing Stage using offline-calculated fingerprints. As in Kopp et al. (2014), the lookup library includes separate GRD fingerprints for seventeen different glacier regions, and thus the spatial pattern associated with glaciers as a whole can change over time in response to the spatial distribution of glacier mass loss.

### 2.3.2.5  Sterodynamic modules

Several modules are included to project sterodynamic sea-level change, i.e., the sum of global-mean thermosteric sea-level rise and ocean dynamic sea-level change (Box 1). As described in Fox-Kemper et al. (2021a), the `tlm/sterodynamics` module does so by taking as input the emulated ocean heat content from the `fair/temperature` module and pre-processed gridded simulations of CMIP6 models. (As noted above, `fair/temperature` is run using a two-layer model representation of the forcing/temperature coupling, from whence comes the abbreviation 'tlm.') Global-mean thermosteric sea-level rise is projected by sampling from a distribution of time-invariant global thermal expansion coefficients derived from CMIP6 simulations (Fox-Kemper et al., 2021a) and multiplying the emulated ocean heat content by those coefficients. The CMIP6 simulations that were used for the calibration of the expansion coefficients are shown in Table A3. The resulting global-mean thermosteric sea-level rise is then combined with ocean dynamic sea-level change and the inverse barometer (IB) effect using the gridded output of CMIP6 models (see the right column of Table A3 for the models that were used in Fox-Kemper et al. (2021a)), based on the time-varying correlation structure between global-mean thermosteric sea-level rise and ocean dynamic sea-level change in the multi-model ensemble. The `tlm/sterodynamics` module expects the CMIP6 input to be pre-processed (e.g., dedrifted and regridded) a priori. The approach used in the provided data set is described in (Fox-Kemper et al., 2021b), and further details are provided by Appendix A.

The sterodynamic component is also provided by the `kopp14/sterodynamics` module, which implements the methodology of Kopp et al. (2014). In this module, drift-corrected global-mean thermosteric sea-level rise is characterized for specific Representative Concentration Pathway scenarios using a t-distribution with the mean and covariance derived from the CMIP5 multi-model ensemble. As in `tlm/sterodynamics`, ocean dynamic sea-level change is then projected using the time-varying correlation structure between global-mean thermosteric sea-level change and ocean dynamic sea-level change in the multi-model ensemble.

As described in Church et al. (2013b), the `ipccar5/thermalexpansion` module projects the distribution of global-mean thermosteric sea-level rise. It is calibrated to the time-dependent mean and standard deviation of the global-mean thermosteric sea-level rise simulated by a multi-model ensemble. Samples are drawn from the mean and standard deviation assuming a normal distribution. The same method was applied by several studies and reports published in between the Fifth and Sixth Assessment Reports of the IPCC (Palmer et al., 2018, 2020; Hermans et al., 2021).

#### 2.3.2.6 Land water storage modules

Two modules provide the land water storage component of sea-level change. As described in Kopp et al. (2014), the first, `kopp14/landwaterstorage`, estimates this component based on the relationship between changes in land water storage and global population change, using United Nations population projections. Reservoir storage is assumed to follow a sigmoidal function of population change, calibrated based on Chao et al. (2008). The relationship between groundwater depletion and population change is based on linear fits to estimates of Wada et al. (2012) and Konikow (2011). The groundwater projection

of Pokhrel et al. (2012), based upon a water resource assessment model, is included as an option for sensitivity analysis. Uncertainty in the projections is generated by sampling the parameters of the sigmoidal fit for reservoir storage and linear fit for groundwater depletion.

The second module, `ssp/landwaterstorage`, follows the methods of Kopp et al. (2014), except for three aspects: (1) instead of using scenario-independent global population projections, population projections of the different SSPs were used

(Samir and Lutz, 2017); (2) the groundwater depletion component was multiplied by 0.8 to account for only 80% of depleted groundwater reaching the ocean (Wada et al., 2016); and (3) the capability to add a temporally linear adjustment for projected reservoir storage based on planned dam construction was added (and applied in AR6 projections using the Hawley et al. (2020) projections for 2020-2040). The GRD fingerprint used is based on the groundwater source pattern of Wada et al. (2012), as described in Slangen et al. (2014).

#### 2.3.2.7 Long-term vertical land motion and glacio-isostatic adjustment modules

Long-term VLM (as well as the sea-surface height contribution from GIA) is provided by `kopp14/verticalandmotion` module. As described in Kopp et al. (2014), this module estimates a constant trend at each spatial location, based upon a Gaussian Process spatio-temporal analysis of tide-gauges in the Permanent Service for Mean Sea Level database. This Gaussian Process analysis modifies an estimated rate of long-term trend derived from a single GIA model (ICE5G-VM2-90), treating

this GIA model as a prior mean rate estimate whose misfits are statistically corrected. Sensitivity tests show this approach exhibits little sensitivity to the choice of initial GIA model in the vicinity of tide gauge records; more substantial differences can occur in parts of the polar region that do not have good observational constraints (Figure A1).

The spatiotemporal model assumes observed RSL can be described as the sum of a uniform (and independently estimated) global component, a regionally-varying, autocorrelated non-linear component (with a decorrelation time scale of order 1-3

years), and a regionally-varying constant trend. The spatial and temporal correlation scales of the regional components are separately tuned (via maximum-likelihood optimization) along different coastal segments. The constant trend is assumed to equal the long-term contribution from VLM (including the VLM term arising from GIA), as well as from the sea-surface height trend arising from GIA, and is propagated into the projection. Uncertainty in the projection is generated based on the uncertainty in the estimate of the constant trend.

Because the statistical model is constructed to extract a century-scale, climate-uncorrelated trend, there should be minimal double-counting of the deformational effects associated with recent land-ice mass loss and land-water redistribution. This

may be a concern along coastlines with only short tide-gauge records, but the resulting bias remains small because future projected rates of land-ice changes are substantially larger than the average rates over the last several decades. VLM associated with future land-ice mass loss and land-water redistribution is incorporated into the GRD projections of those components'
respective modules.

An alternative, direct-sampling-based VLM approach is demonstrated by the `NZInsarGPS/verticallandmotion` module, which reads and samples gridded land motion data described in an external file and extrapolates these rates linearly into the future. In Naish et al. (in review), this module applies a gridded data file describing rates of land motion inferred from interferometric synthetic aperture radar (InSAR) data.

### 330   2.3.3   Totaling module

The `facts/total` module handles the aggregation of sea-level component probability distributions into probability distributions for total GMSL and RSL change. This module takes as an input a configuration file pointing to the output files that constitute different Workflows (see Section 2.4).

### 2.3.4   ESL module

The `extremesealevel/pointsoverthreshold` module, which is based on the methods of Oppenheimer et al. (2019a) and Frederikse et al. (2020), first derives declustered ESLs from tide gauge data from the GESLA2 database (Woodworth et al., 2016) using a peak-over-threshold method with a user-defined threshold percentile. After removal of the annual means, a Generalized Pareto Distribution is fitted to the declustered extremes using maximum likelihood estimation. The estimated parameters and their uncertainty are used to generate ESL return-period curves. Below the threshold of the Generalized Pareto
Distribution, a Gumbel distribution with support between Mean Higher High Water and the threshold is assumed and used to compute return periods, following (Buchanan et al., 2016). In the projection Stage, the module augments the return-period curves by projected RSL change to project how the expected frequency of ESL events of different magnitudes change as the baseline height of the events is increased (Frederikse et al., 2020). Note that this approach assumes that the ESL distribution, relative to a changing mean sea level, is stationary; it does not account for factors such as changes to storm frequency, intensity,
or tracks.

### 2.4   Workflows

In this paper, we demonstrate FACTS' capabilities by implementing seven different Workflows (i.e., sets of sea-level component modules) (Table 2). The Workflows align with those implemented by IPCC AR6. As previously described in the description of the IPCC AR6 land-ice modules, in FACTS 1.0, we replace the direct-sampling of offline calculated values
used in AR6 with GSAT-driven `emulandice` and `larmip` modules. The Workflows share a common set of modules used for projecting VLM (`kopp14/verticallandmotion`), sterodynamic sea level (`tlm/sterodynamics`), and land water storage (`ssp/landwaterstorage`). They differ based on their handling of the cryospheric components (ice sheets and glaciers).

**Table 1.** Modules included in FACTS 1.0

| Category | Module | Drivers |
|---|---|---|
| Climate | `fair/temperature` | emissions |
| Generic sea-level component | `facts/directsample` | static |
| Glaciers | `emulandice/glaciers` | temperature |
| Glaciers | `kopp14/glaciers` | static by RCP scenario |
| Glaciers | `ipccar5/glaciers` | temperature |
| Glaciers | `ipccar6/gmipemuglaciers` (deprecated) | static by SSP scenario |
| Antarctic and Greenland Ice Sheets | `bamber19/icesheets` | static by warming level scenario |
| Antarctic and Greenland Ice Sheets | `ipccar5/icesheets` | temperature |
| Antarctic and Greenland Ice Sheets | `ipccar6/ismipemuicesheets` (deprecated) | static by SSP scenario |
| Antarctic and Greenland Ice Sheets | `kopp14/icesheets` | static by RCP scenario |
| Antarctic Ice Sheet | `deconto21/AIS` | static by RCP scenario |
| Antarctic Ice Sheet | `emulandice/AIS` | temperature |
| Antarctic Ice Sheet | `ipccar6/larmipAIS` (deprecated) | static by SSP scenario |
| Antarctic Ice Sheet | `larmip/AIS` | temperature |
| Greenland Ice Sheet | `emulandice/GrIS` | temperature |
| Greenland Ice Sheet | `FittedISMIP/GrIS` | temperature |
| Land Water Storage | `kopp14/landwaterstorage` | static |
| Land Water Storage | `ssp/landwaterstorage` | population |
| Sterodynamic Sea Level | `kopp14/sterodynamics` | static by RCP scenario |
| Sterodynamic Sea Level | `ipccar5/thermalexpansion` | static by RCP scenario |
| Sterodynamic Sea Level | `tlm/sterodynamics` | ocean heat content for global-mean projection local correlation by SSP scenario |
| Vertical land motion | `kopp14/verticallandmotion` | static |
| Vertical land motion | `NZInsarGPS/verticallandmotion` | static |
| Integration | `facts/total` | sea-level components |
| Extreme sea level | `extremesealevel/pointsoverthreshold` | total relative sea level |

The `ipccar6` modules are direct-sample modules that were used only in IPCC AR6, and have been deprecated in FACTS 1.0 in favor of the `emulandice` and `larmip` modules. The `ipccar5` modules indicate the methods described in Church et al. (2013b), which in some cases and contexts were used by AR6, as described in Fox-Kemper et al. (2021a) and Table 2. The `ipccar5/glaciers` module includes, in addition to the original IPCC Fifth Assessment Report calibration, recalibrations to GlacierMIP and GlacierMIP2 (Hock et al., 2019; Marzeion et al., 2020). The GlacierMIP2 recalibration is used in IPCC AR6 and in this paper and is denoted by a parenthetical '(GMIP2)' in Tables 2 and 3.

Workflows 1e and 2e employ Gaussian Process emulation of ice sheet and glacier intercomparison exercise outputs for Greenland and glaciers and, in the case of Workflow 1e, Antarctica (i.e., `emulandice` in Table 2). However, the Gaussian 355 Process emulator of Edwards et al. (2021) models each time point independently, and thus does not estimate rates. Because

**Table 2.** Workflows used in this paper

| Workflow | GrIS | AIS | Glaciers | Land Water | Sterodynamic | VLM |
|---|---|---|---|---|---|---|
| *Medium confidence* workflows | | | | | | |
| 1e | `emulandice` | `emulandice` | `emulandice` | `ssp` | `tlm` | `kopp14` |
| 1f | `FittedISMIP` | `ipccar5` | `ipccar5` (**GMIP2**) | `ssp` | `tlm` | `kopp14` |
| 2e | `emulandice` | `larmip` | `emulandice` | `ssp` | `tlm` | `kopp14` |
| 2f | `FittedISMIP` | `larmip` | `ipccar5` (**GMIP2**) | `ssp` | `tlm` | `kopp14` |
| | | | | | | |
| *Low confidence* workflows | | | | | | |
| 3e | `emulandice` | `deconto21` | `emulandice` | `ssp` | `tlm` | `kopp14` |
| 3f | `FittedISMIP` | `deconto21` | `ipccar5` (**GMIP2**) | `ssp` | `tlm` | `kopp14` |
| 4 | `bamber19` | `bamber19` | `ipccar5` (**GMIP2**) | `ssp` | `tlm` | `kopp14` |

Workflows used in this paper match those of AR6 (Fox-Kemper et al., 2021a), except that in AR6, results from ISMIP/GlaicerMIP emulation and
LARMIP-2 were computed offline by those models' authors and then added into the projections as static data, rather than online in a FACTS experiment
as done by the `emulandice` and `larmip` modules.

`emulandice` uses a non-parametric (Gaussian process) model, where no functional form is assumed, rather than a parametric model, in which dependencies are asserted, Workflows using `emulandice` modules can only project up to the end of the original simulations (rather than extrapolate beyond them) and therefore end in 2100. Workflows 1f and 2f therefore substitute alternative, parametric representations for GrIS and glaciers. Workflows 2e and 2f differ from Workflows 1e and 1f by

employing an alternative Antarctic ice sheet emulator, provided by the `larmip` module. These four Workflows together form the basis of the *medium confidence* projections presented by AR6 (Fox-Kemper et al., 2021a) (for example, in the unshaded columns of Table 9.9 of Fox-Kemper et al. (2021a)). Workflows 3e, 3f and 4, by contrast, take alternative approaches to ice sheet representation intended to capture processes not represented in most ice-sheet models. Workflows 3e and 3f employs the `deconto21` projections for Antarctica, while Workflow 4 employs structured expert judgement-based projections (`bamber19`)

for both Greenland and Antarctic ice sheets. These three Workflows are combined with the *medium confidence* Workflows to form the basis of the broader AR6 *low confidence* projections (for example, for SSP5-8.5, in the final column of Table 9.9 of Fox-Kemper et al. (2021a)).

## 3   Results

All results presented are based on 2000 pseudo-random Monte Carlo samples. To illustrate the application of FACTS, we focus

on GMSL projections and on RSL and ESL projections at a single site, New York City.

### 3.1 Temperature projections

FACTS experiments begin with the estimation of the GSAT response to emissions forcing, as projected by the FaIR climate emulator. By construction, these projections are generally consistent with those of AR6 (Lee et al., 2021), with median warming in 2100 above 1850-1900 of 1.6°C in SSP1-2.6, 2.6°C in SSP2-4.5, and 4.7°C in SSP5-8.5 (Table 3). Note that SSP1-2.6 is aligned with the Paris Agreement goal of limiting warming to well-below 2 C, while SSP2-4.5 is closer to projected emissions under current policy. SSP5-8.5 emissions represent a high-end trajectory that would require a reversion to fossil-fuel-intensive development (Riahi et al., 2022).

### 3.2 Global-mean contributions from sea-level components

In the sea-level component Experiment Step, FACTS estimates the contributions to future GMSL and RSL rise from the cryosphere, land water storage, and sterodynamic sea-level change. Some sea-level components modules (for example, the sterodynamic, ice sheet, and glacier modules used in workflows 1e, 1f, 2e, and 2f) take the FaIR-projected warming as an input. Others rely upon pre-computed projections, in some cases indexed by SSP or RCP emissions scenario (for example, the `deconto21` and `bamber19` ice sheet modules, and the deprecated `ipccar6` ice sheet and glacier modules) (Table 1).

Projected median and 17th-83rd percentile GMSL contributions are shown in Table 3 and Figure 2. The cryosphere as a whole (including glaciers and polar ice sheets) dominates median projections for 2100 under all emissions scenarios, but the relative contribution of polar ice sheets in particular varies substantially across modules. This is particularly the case under very high emissions (SSP5-8.5/RCP8.5), where one module (`deconto21`) projects the Antarctic contribution to be the single largest term. For the polar ice sheet contributions, GMSL contributions projected by different modules are similar until 2040 but begin to diverge beyond 2050, and this divergence is larger for higher emission scenarios (Figure 2). By contrast, both glacier modules (`ipccar5` and `emulandice`) remain consistent throughout this century; this is to be expected, given that both are calibrated to the same underlying GlacierMIP ensemble of glacier model projections (Marzeion et al., 2020).

### 3.3 Total global-mean sea-level change projections

Total GMSL projections (Table 4, Figure 3) are generally in close agreement between Workflows using the `emulandice` emulators of ISMIP6 and GlacierMIP projections (i.e., Workflows 1e, 2e, and 3e) and the corresponding Workflows that substitute parametric emulators (i.e., Workflows 1f, 2f, and 3f) (Table 4). For example, under SSP2-4.5, total projections are 0.50 (0.42–0.60, 17th–83rd percentile range) m under Workflow 1e and 0.49 (0.40–0.59) m under Workflow 1f; differences are smaller for lower emissions scenarios and for other `emulandice`/parametric Workflow pairs (i.e., 2e vs. 2f, and 3e vs. 3f), and larger for higher emissions. For the remainder of this paper, we therefore focus primarily on Workflows 1f, 2f, 3f, and 4. The text discusses primarily SSP5-8.5, for which different Workflows show the greatest distinctions. Figures highlight the difference between SSP1-2.6 and SSP5-8.5, as these are the two scenarios that can be projected using all Workflows.

Substantial differences arise between Workflows based particularly on the choice of Antarctic module. Under SSP5-8.5, median projections for 2100 differ by 0.14 m between Workflow 1f (Antarctica calibrated as per IPCC AR5: 0.66 (0.55-0.78)

m) and Workflow 2f (Antarctica calibrated to LARMIP2: 0.80 (0.60-1.00) m), with the latter projections also exhibiting fatter tails. This reflects the differences seen at the component level (Table 3, Figure 2). Larger differences are seen under higher

emissions scenarios with the two Workflows that AR6 employed to incorporate *low confidence* processes. Both Workflow 3f (Antarctic ice sheet modeling MICI: 0.97 (0.81-1.17) m) and Workflow 4 (both Antarctica and Greenland based on structured expert judgement: 1.00 (0.69-1.64) m) have median projections for SSP5-8.5 exceeding those of the *medium confidence* Workflows by at least 0.17–0.20 m. The median projections for Workflow 3e and 4 are closely aligned, but the structured expert judgment-based projections (Workflow 4) span a larger range, reflecting primarily greater Greenland ice sheet uncertainty than

in Workflow 3f.

Consistent with these observations, by the end of the century, total projection variance is generally dominated by polar ice sheet uncertainty, particularly under Workflows 2f, 3f and 4 (Figure 4). In addition, Workflows 1f and 3f reveal a positive interaction term: i.e., the variance of GMSL projections is greater than the sum of the variances of the individual components. This positive interaction term arises because global-mean thermosteric sea-level rise, glacier loss, and (in the *medium confidence*

Workflows) polar ice sheet loss share a common dependence on GSAT and thus are positively correlated.

## 3.4 Relative and extreme sea-level projections at New York City

The differences between projected GMSL rise and projected RSL rise at New York City are consistent with past studies (e.g., Kopp et al., 2014) (Table 4; Figures 5, 6, 7). The median contribution and variance arising from the distant Antarctic is increased due to GRD effects, which cause West Antarctic Ice Sheet loss to cause about 20% greater sea-level rise at New York

City than in the global mean; while the median contributions and variance arising from the Greenland Ice Sheet and global glaciers are reduced due to relative proximity. The median sterodynamic contribution and its variance are larger than global-mean thermosteric sea-level rise due to the potential contribution from a slowdown of the Atlantic Meridional Overturning Circulation (Yin et al., 2009; Yin and Goddard, 2013). A long-term GIA trend, arising primarily from land subsidence, adds a steady $1.5 \pm 0.2$ mm/yr to RSL rise, shifting all projections upward but contributing little to variance.

As with GMSL projections, substantial differences arise between Workflows based particularly on the choice of Antarctic module. Under SSP5-8.5, Workflow 1f (0.90 [0.71–1.10] m) and Workflow 2f (1.07 [0.86–1.34] m) differ by 0.17 m in the median. Further differences are seen with the two Workflows that AR6 employed to incorporate *low confidence* processes. Notably, because high-end GMSL projections in Workflow 4 include a larger Greenland contribution than in other Workflows, and because Greenland's effects on RSL rise at New York City are less than in GMSL rise, median Workflow 4 RSL projections

(1.22 [0.88–1.73] m) are lower than Workflow 3e (1.27 [1.04–1.51] m), which relies more heavily on Antarctica to drive high-end GMSL rise. While the Workflow 4 tail remains the fattest of all Workflows, Workflow 3e's tail is fattened substantially as compared to GMSL because of the heightened response of New York City RSL to Antarctic mass loss.

Differences in RSL projections translate into differences in ESL projections (Table 5, Figure 8). For example, under Workflow 1f, the historic 1% average annual probability extreme sea level at New York City (estimated at 1.83 m above Mean

Higher High Water) is projected to occur 2.6 (1.8–4.2) times more often by 2050 and 6.5 (3.2–17.2) times more often by 2100 under SSP1-2.6 due to the effects of RSL rise, and 2.8 (1.9–4.7) times more often by 2050 and 22.1 (8.0–90.3) times more

**Table 3.** Component Projections for 2100

| Component | Module | SSP1-2.6 | SSP2-4.5 | SSP5-8.5 |
|---|---|---|---|---|
| GSAT (°C) | `fair/temperature` | 1.63 (1.35–1.99) | 2.61 (2.19–3.12) | 4.66 (3.96–5.55) |
| Glaciers | `emulandice/glaciers` | 0.09 (0.07–0.11) | 0.12 (0.10–0.14) | 0.18 (0.15–0.20) |
| Glaciers | `ipccar5/glaciers` (GMIP2) | 0.09 (0.06–0.13) | 0.12 (0.08–0.16) | 0.16 (0.11–0.22) |
| Glaciers | `kopp14/glaciers*` | 0.11 (0.08–0.14) | 0.13 (0.09–0.16) | 0.17 (0.13–0.21) |
| Antarctica | `bamber19/icesheets` | 0.10 (-0.01–0.26) | — | 0.20 (0.02–0.57) |
| Antarctica | `deconto21/AIS` | 0.08 (0.06–0.11) | 0.09 (0.07–0.11) | 0.34 (0.19–0.53) |
| Antarctica | `emulandice/AIS` | 0.08 (0.03–0.14) | 0.08 (0.03–0.14) | 0.08 (0.03–0.14) |
| Antarctica | `ipccar5/icesheets` | 0.06 (-0.01–0.14) | 0.05 (-0.02–0.13) | 0.04 (-0.04–0.11) |
| Antarctica | `kopp14/icesheets*` | 0.06 (-0.05–0.16) | 0.05 (-0.06–0.16) | 0.04 (-0.08–0.14) |
| Antarctica | `larmip/AIS` | 0.13 (0.05–0.26) | 0.14 (0.05–0.29) | 0.15 (0.05–0.34) |
| Greenland | `bamber19/icesheets` | 0.13 (0.07–0.30) | — | 0.22 (0.10–0.59) |
| Greenland | `emulandice/GrIS` | 0.05 (0.01–0.10) | 0.08 (0.04–0.13) | 0.12 (0.08–0.18) |
| Greenland | `FittedISMIP/GrIS` | 0.08 (0.06–0.10) | 0.10 (0.08–0.12) | 0.14 (0.11–0.18) |
| Greenland | `ipccar5/icesheets*` | 0.08 (0.05–0.10) | 0.09 (0.07–0.13) | 0.16 (0.11–0.22) |
| Greenland | `kopp14/icesheets*` | 0.06 (0.03–0.11) | 0.08 (0.03–0.15) | 0.14 (0.07–0.25) |
| Land Water Storage | `kopp14/landwaterstorage` | 0.05 (0.03–0.07) | 0.05 (0.03–0.07) | 0.05 (0.03–0.07) |
| Land Water Storage | `ssp/landwaterstorage` | 0.03 (0.02–0.04) | 0.03 (0.02–0.04) | 0.03 (0.02–0.04) |
| Thermal Expansion | `ipccar5/thermalexpansion*` | 0.15 (0.13–0.18) | 0.21 (0.18–0.23) | 0.32 (0.28–0.36) |
| Thermal Expansion | `tlm/sterodynamics` | 0.14 (0.11–0.17) | 0.19 (0.15–0.23) | 0.29 (0.24–0.35) |

Median (17th-83rd percentile) projections produced by FACTS modules. All components except GSAT are in m GMSL contribution relative to a 1995-2014 baseline. Global-mean surface air temperature (GSAT) is in °C relative to a 1850–1900 baseline. For certain modules (marked with asterisk), projections for Representative Concentration Pathways 2.6, 4.5, and 8.5 are shown in lieu of those for the SSP scenarios.

often by 2100 under SSP5-8.5. 83rd percentile projected amplification factors are all $< 6.8$ by 2100 under SSP1-2.6, but under SSP5-8.5 and Workflow 4, the 83rd percentile SSP5-8.5 amplification factor exceeds 10,000 – meaning that, under the high end of this fat-tailed projection, the historic 100-year ESL event might occur over 100 times per year. (Note that ESL return periods do not translate directly into flooding or flood damages; see Rasmussen et al. (2022) for a critique of ESL amplification factors as a metric and Hermans et al. (2023) for presentation of a related approach.)

**Table 4.** Total Projections for 2100

| Workflow | SSP1-1.9 | SSP1-2.6 | SSP2-4.5 | SSP3-7.0 | SSP5-8.5 |
|---|---|---|---|---|---|
| Global-mean sea-level change (meters) | | | | | |
| 1e | 0.35 (0.27–0.44) | 0.40 (0.32–0.49) | 0.50 (0.42–0.60) | 0.62 (0.53–0.73) | 0.71 (0.61–0.82) |
| 1f | 0.35 (0.27–0.44) | 0.40 (0.31–0.49) | 0.49 (0.40–0.59) | 0.58 (0.48–0.68) | 0.66 (0.55–0.78) |
| 2e | 0.40 (0.30–0.53) | 0.46 (0.35–0.60) | 0.57 (0.45–0.73) | 0.70 (0.57–0.88) | 0.80 (0.65–1.00) |
| 2f | 0.41 (0.33–0.54) | 0.48 (0.38–0.62) | 0.59 (0.48–0.74) | 0.70 (0.58–0.87) | 0.80 (0.66–1.00) |
| 3e | — | 0.40 (0.34–0.48) | 0.51 (0.45–0.59) | — | 0.97 (0.80–1.18) |
| 3f | — | 0.43 (0.37–0.49) | 0.53 (0.47–0.61) | — | 0.97 (0.81–1.17) |
| 4 | — | 0.53 (0.37–0.80) | — | — | 1.01 (0.69–1.64) |
| Relative sea-level change at New York City (meters) | | | | | |
| 1e | 0.56 (0.36–0.79) | 0.62 (0.45–0.79) | 0.75 (0.58–0.93) | 0.86 (0.69–1.04) | 0.97 (0.79–1.15) |
| 1f | 0.55 (0.33–0.77) | 0.60 (0.42–0.78) | 0.72 (0.54–0.90) | 0.81 (0.62–0.99) | 0.90 (0.71–1.10) |
| 2e | 0.64 (0.41–0.88) | 0.70 (0.50–0.91) | 0.85 (0.65–1.07) | 0.97 (0.76–1.21) | 1.09 (0.86–1.35) |
| 2f | 0.64 (0.41–0.89) | 0.70 (0.51–0.92) | 0.84 (0.65–1.07) | 0.95 (0.74–1.19) | 1.07 (0.86–1.34) |
| 3e | — | 0.63 (0.47–0.80) | 0.77 (0.62–0.94) | — | 1.27 (1.04–1.51) |
| 3f | — | 0.64 (0.48–0.81) | 0.78 (0.62–0.94) | — | 1.26 (1.03–1.51) |
| 4 | — | 0.71 (0.48–0.97) | — | — | 1.22 (0.89–1.73) |

Median (17th-83rd percentile) projections are shown relative to a 1995-2014 baseline.

**Table 5.** Frequency amplification factors in the years 2050 and 2100 for the historic 1% average annual probability (100-year return period) extreme sea-level event at New York City

| Workflow | 2050 | | 2100 | |
|---|---|---|---|---|
| | SSP1-2.6 | SSP5-8.5 | SSP1-2.6 | SSP5-8.5 |
| 1f | 2.6 (1.8–4.2) | 2.8 (1.9–4.7) | 6.5 (3.2–17.2) | 22.1 (8.0–90.3) |
| 2f | 2.9 (1.9–4.8) | 3.2 (2.1–5.4) | 10.0 (4.1–33.8) | 55.7 (13.0–451.2) |
| 3f | 2.7 (1.9–4.3) | 2.9 (2.0–4.7) | 7.6 (3.7–19.6) | 146.5 (27.6–1629.2) |
| 4 | 2.9 (1.9–5.2) | 3.4 (2.1–6.8) | 9.9 (4.0–43.6) | 126.4 (16.3–10,283.4) |

Median (17th-83rd percentile) projections are shown, as ratio of event probability in 2050 or 2100 to event probability in 1995–2014.

# 4 Discussion

## 4.1 Applications to date

The modular approach adopted by FACTS intentionally lends itself to careful consideration of both parametric and structural
uncertainty in sea-level projections. Indeed, FACTS modules have already been used to support several major assessments of

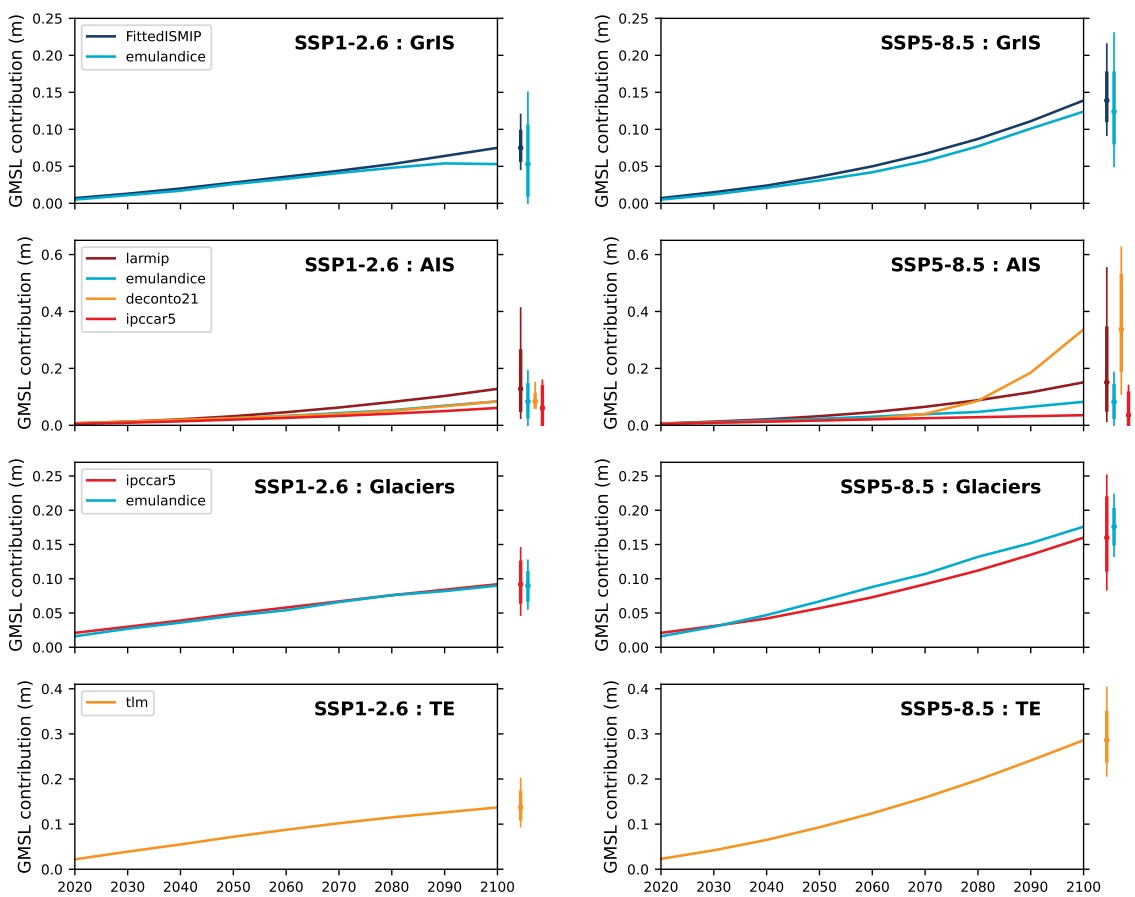

**Figure 2.** GMSL contributions from the Greenland ice sheet (GrIS), the Antarctic ice sheet (AIS), glaciers, and thermal expansion (TE) for SSP1-2.6 and SSP5-8.5, based upon different FACTS modules. Curves show median projections. Thick/thin bars at right show 17-83rd/5-95th percentile projections for 2100.

sea-level change. As previously noted, the IPCC AR6 sea-level projections were developed using FACTS modules, and the example Workflows described in this paper replicate the AR6 analysis within rounding errors (Table A1). Slightly larger discrepancies with total projections (Table A2) are attributable to the combination of rounding errors and differences in sampling. (Note that AR6 used 20,000 samples per workflow, compared to the 2,000 per workflow in the results shown here.)

AR6 followed the development of Workflow probability distributions with a particular approach to combine alternative probability distributions based upon probability boxes, or p-boxes (Kriegler and Held, 2005; Le Cozannet et al., 2017). P-boxes describing a set of probability distributions encompass the cumulative distribution functions of the underlying probabilities; for example, the outer 17th-83rd percentile range of a p-box spans from the lowest 17th percentile of all distributions considered to the highest 83rd-percentile. All the distributions considered by construction agree that there is *at least* a 66% chance that

the true value falls within this particular range. Fox-Kemper et al. (2021a) used outer 17th–83rd percentile p-box ranges to

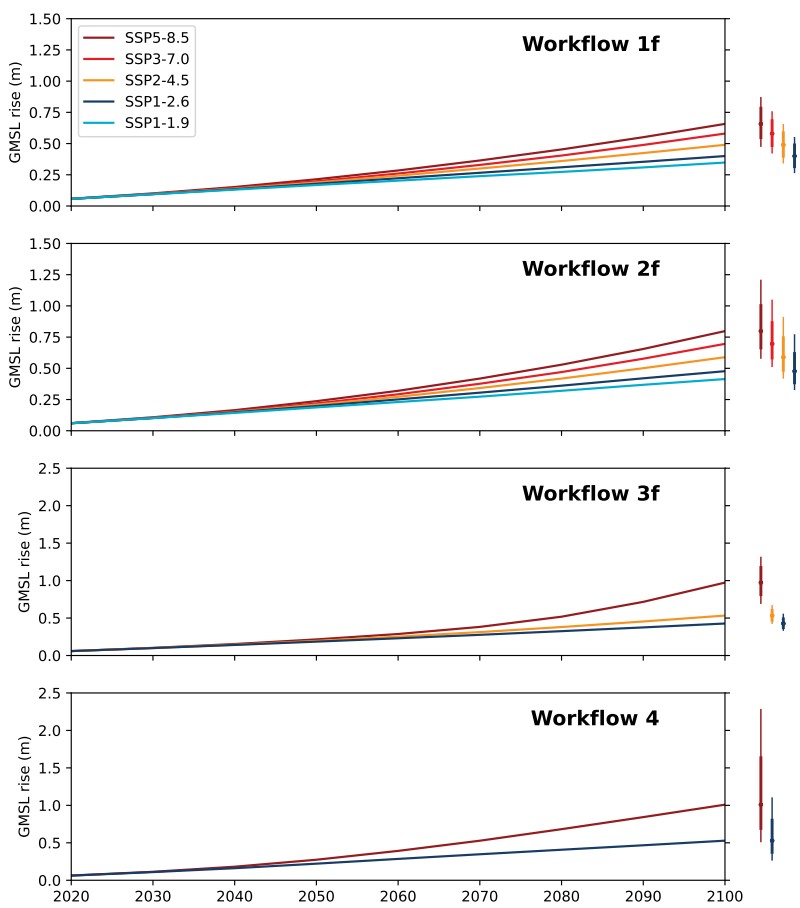

**Figure 3.** Total GMSL projections under four different Workflows under different SSP scenarios. Curves show median projections. Thick/thin bars at right show 17-83rd/5-95th percentile projections for 2150. Note change of y-axis scale between Workflows 1f and 2f (top two rows), representing *medium confidence* processes, and Workflows 3e and 4, which include *low confidence* processes.

characterize its *likely* ranges, where *likely* in IPCC terminology means a 66–100% chance (Mastrandrea et al., 2010). (This is a difference from the definition of *likely range* used in the rest of the IPCC AR6 Working Group 1 report, which specifically refers to the 17th–83rd percentile of a single estimated probability distribution.) Workflows employing ISMIP and GlacierMIP emulators (1e, 2e, and 3e) were preferred over those with simple parametric representations for land ice where possible, but
Workflows employing these simple representations (1f, 2f and 3f) were used when required for rates, which were not emulated. Workflows 1e/1f and 2e/2f were combined in a p-box to produce the AR6 *medium confidence* projections, while Workflows 3e/3f and 4 were added for *low confidence* projections.

The US Interagency Task Force on Sea-level Rise Scenarios (Sweet et al., 2022) built upon the same FACTS output as AR6, but took a different approach to summarizing their results. Intending to produce a set of plausible global and regional sea-level
scenarios to guide decision making – rather than, as in AR6, to characterize the likelihood of different future outcomes – the

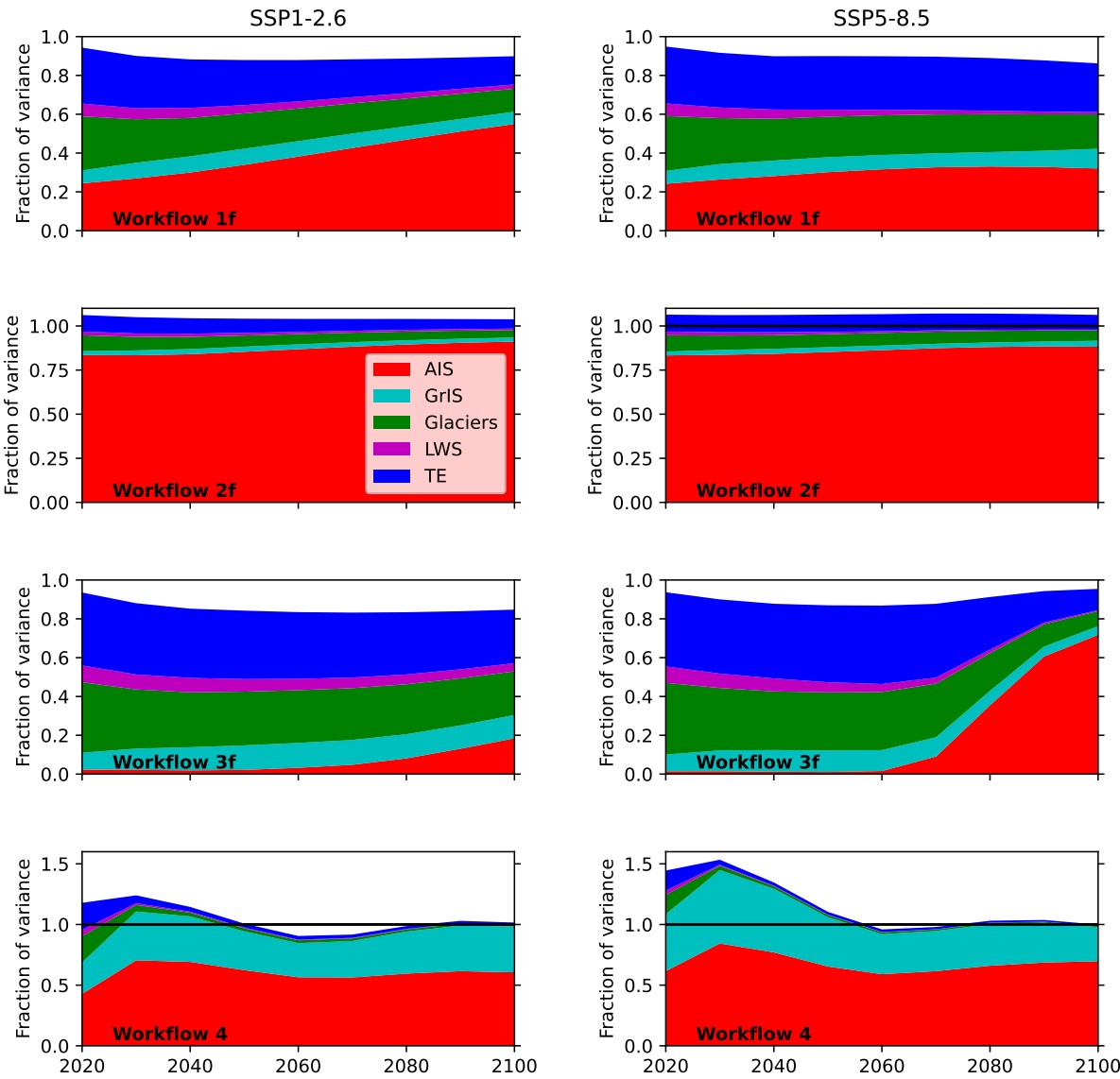

**Figure 4.** Variance decomposition of GMSL change through 2100 under SSP1-2.6 (left column) and SSP5-8.5 (right column), under Workflows 1f, 2f, 3f and 4 (top to bottom), in the style of Hawkins and Sutton (2009). Each colored wedge represents the variance across Monte Carlo samples for a particular component, under the specified scenario and Workflow, normalized by the variance of projections for total sea-level change in the same scenario and Workflow. The difference between the sum of component variances and the total variance (normalized to 1.0) represents the interaction among components.

Task Force filtered the samples of sea-level rise associated with Workflows 1f, 2f, 3f and 4 to identify five subsets consistent with a range of 21st century GMSL rise. This range was semi-independently defined, based in part on an interpretation of the

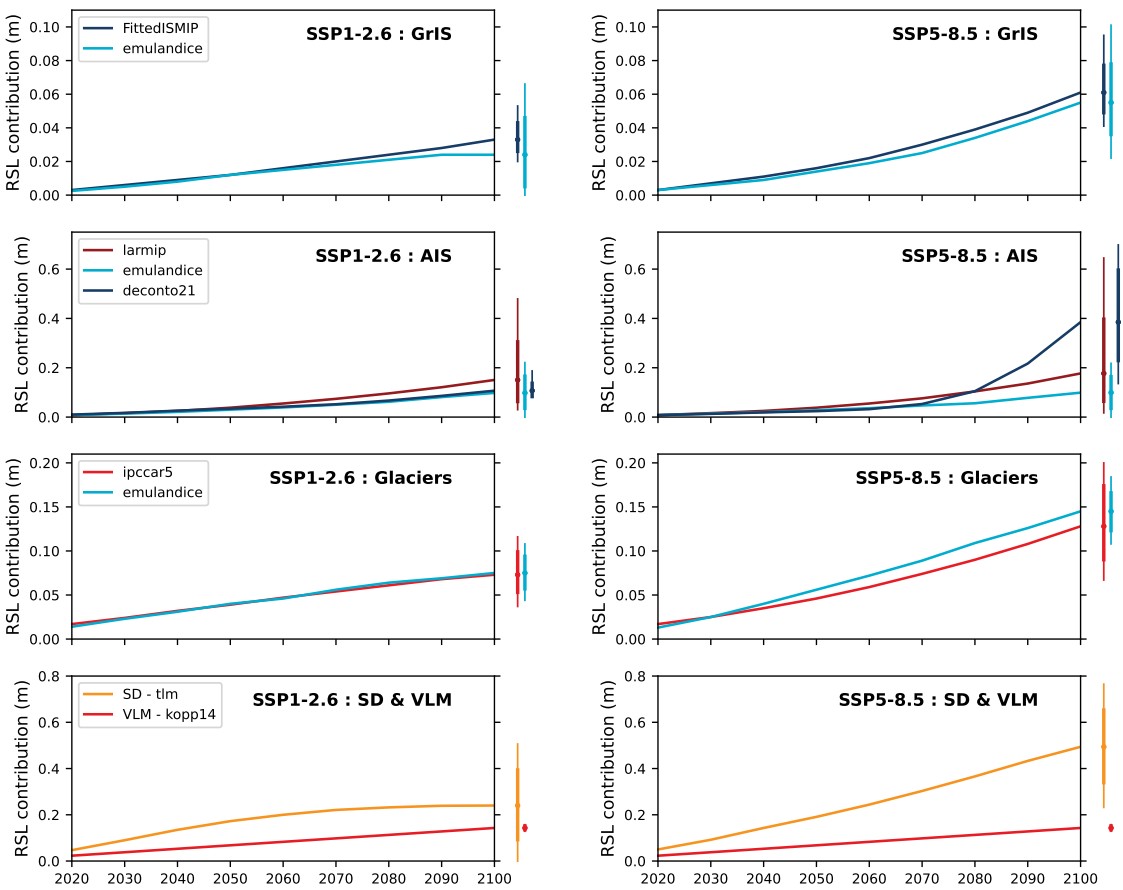

**Figure 5.** Projected New York City RSL contributions from the Greenland ice sheet, the Antarctic ice sheet, glaciers, and sterodynamic sea level from different FACTS modules under SSP1-2.6 and SSP5-8.5. Curves show median projections. Thick/thin bars at right show 17-83rd/5-95th percentile projections for 2100.

range of values presented in the AR6, to span from as low as 0.3 m (roughly, a continuation of late 20th century GMSL range) to as high as 2.0 m, the latter informed by AR6's conclusion that low-likelihood, high-impact processes could elevate GMSL
above the *likely* range by more than one metre. The median of each subset forms the center of each set of GMSL and RSL scenarios, while the 17th and 83rd percentiles of each subset provide within-scenario high/low sensitivity cases.

Both IPCC and the US Interagency Task Force invested significant effort in communicating these projections. For example, the NASA Sea Level Change Team, in partnership with these two groups, developed interactive projection viewers (at https://sealevel.nasa.gov/ipcc and https://sealevel.nasa.gov/task-force-scenario-tool) to allow practitioners to explore the projections
for sites around the world and the US, respectively.

FACTS has also been used to develop national RSL projections for New Zealand (Levy et al., 2020; Naish et al., in review). In these studies, existing Workflows (either based on Kopp et al. (2014) or matching those employed in AR6) were

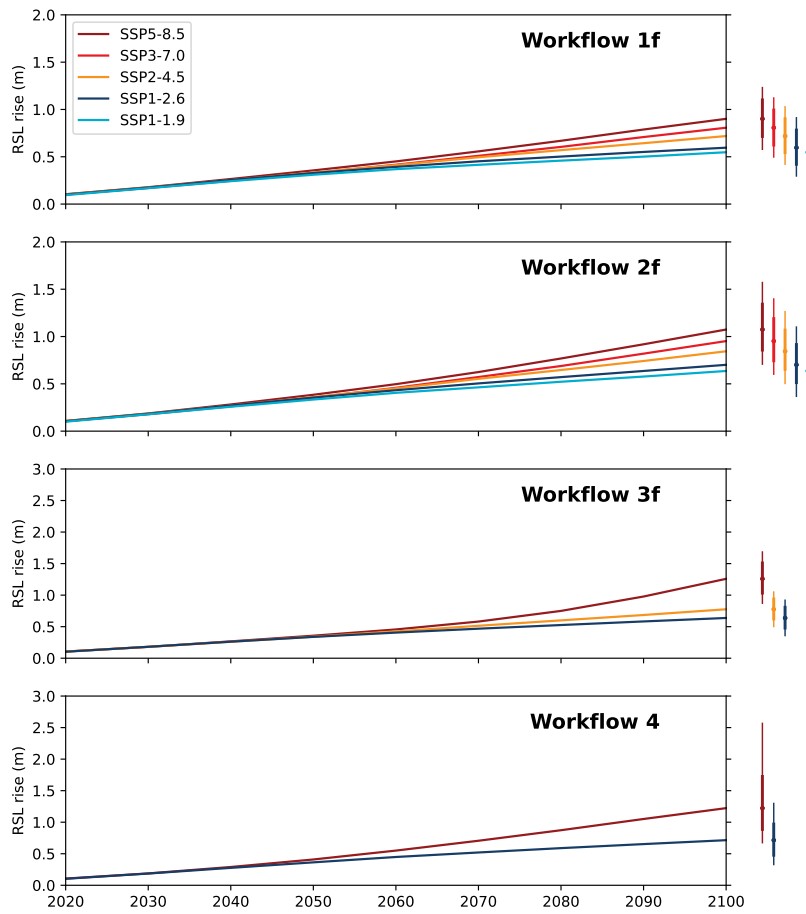

**Figure 6.** Total RSL projections for New York City under four different Workflows under different SSP scenarios. Curves show median projections. Thick/thin bars at right show 17-83rd/5-95th percentile projections for 2150.

amended by replacing the existing Kopp et al. (2014)-based projections of GIA and VLM with gridded estimates based on interferometric synthetic aperture radar (InSAR) data, calibrated with ground-based Global Navigation Satellite System (GNSS) measurements. This substitution reflects a need common to many national and subnational sea-level assessments, which seek consistency with broader assessments while substituting in information that can be assessed in greater detail at a local scale.

## 4.2 Directions for improvement

From a scientific perspective, a number of different directions promise improvement in FACTS projections.

At present, many but not all the modules within FACTS accept climate information as an input. In particular, the ice-sheet modules used to project deeply uncertain ice-sheet processes (the `bamber19/icesheets` and `deconto21/AIS` modules) rely upon direct sampling of output generated by individual studies (Bamber et al., 2019; DeConto et al., 2021). This means they

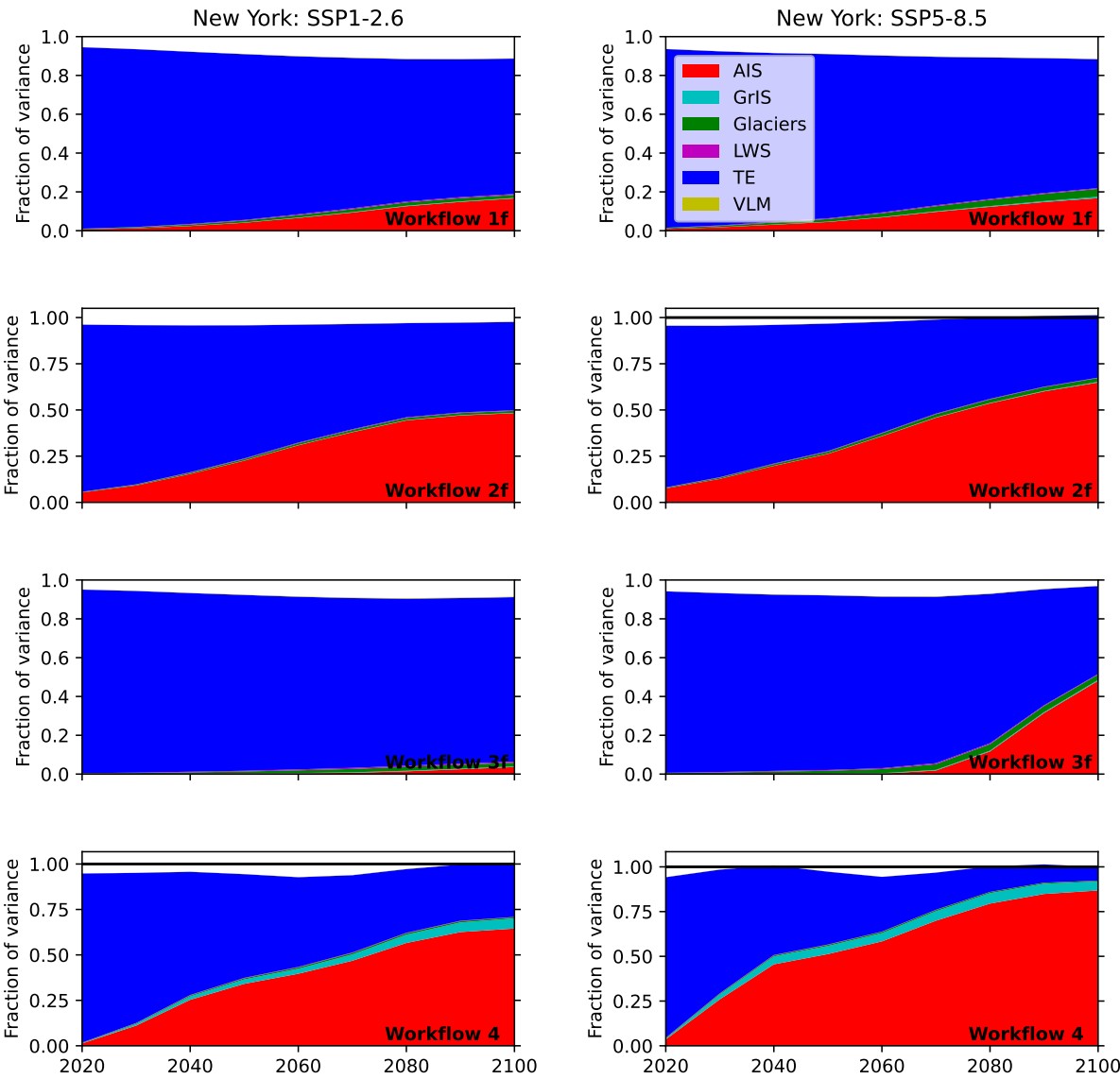

**Figure 7.** Variance decomposition of New York City RSL change through 2100 under SSP1-2.6 (left column) and SSP5-8.5 (right column), under Workflows 1f, 2f, 3f and 4 (top to bottom), in the style of Hawkins and Sutton (2009). Each colored wedge represents the variance across Monte Carlo samples for a particular component, under the specified scenario and Workflow, normalized by the variance of projections for total sea-level change in the same scenario and Workflow. The difference between the sum of component variances and the total variance (normalized to 1.0) represents the interaction among components.

can be applied only to a limited set of climate scenarios. For example, Bamber et al. (2019) produced projections for 2°C and 5°C GSAT stabilization scenarios. With some caveats, these are used by AR6 to inform the projections for SSP1-2.6 and

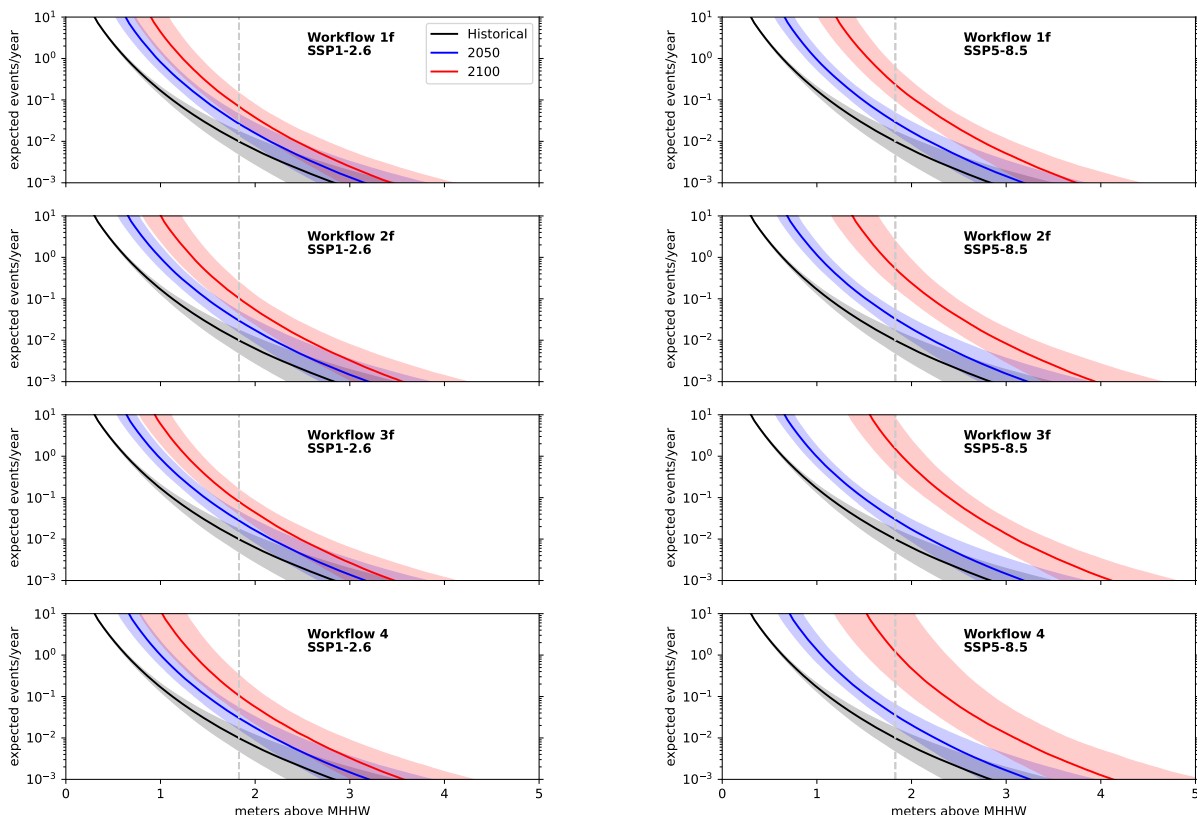

**Figure 8.** Extreme sea-level return-period curves at New York City under SSP1-2.6 and SSP5-8.5, under Workflows 1f, 2f, 3f and 4, in the historical period (black) and the years 2050 (blue) and 2100 (red). Solid line shows median projection; shading shows 17th-83rd percentile estimates. Dashed vertical line indicates the extreme sea-level height associated with the historic 1% average annual probability event.

SSP5-8.5 (though SSP1-2.6 most likely stabilizes below 2°C and SSP5-8.5 continues after 2100 to warm well above 5°C).
Generalizing emulation approaches to encompasses these alternative sources of information would allow projections of *low confidence* processes to be generated for arbitrary climate scenarios; doing this cautiously might require either advances in the primary literature or a great deal of humility and uncertainty regarding the assumptions used for scenario interpolation. (It is difficult, for example, to infer the warming level associated with critical thresholds in ice sheet behavior from only two climate scenarios.)

The existing VLM modules assume a constant-rate trend into the future. While perhaps the best assumption that can be undertaken at a global scale, more refined approaches might be possible at a local scale. For example, in many regions, VLM is driven in part by highly-localised subsidence associated with anthropogenic interventions, such as fluid withdrawal and/or surface loading (Shirzaei et al., 2021). Indeed, in the regions of the world experiencing the fastest rates of RSL rise, these currently tend to be the largest drivers. In such cases, assumption of a constant-rate trend in future may not be the most suitable
assumption. A module capable of representing alternative scenarios of such factors and their evolution over time could be

helpful in assessments in such regions (e.g., Minderhoud et al., 2020). Alternatively, such scenarios might be independently added to RSL projections that have the VLM component removed.

The current VLM modules also do not explicitly address uncertainty in GIA (e.g., Melini and Spada, 2019). In the `kopp14/verticallan` module, GIA uncertainty does not make a substantial contribution in locations where tide-gauges are available to constrain long-term changes; however, this uncertainty can be significant at sites distant from tide-gauges, particularly in polar regions where the GIA contribution is largest (Figure A1). New approaches to fusing model projections with geological, tide-gauge, and satellite observations could better characterize this uncertainty (e.g., Caron et al., 2018).

The existing ESL module treats the shape and scale of ESL return period curves as stationary, with the distribution only shifted vertically by the increment of RSL change. In fact, ESL return periods will change due to processes such as shifts in tropical cyclone intensity and tracks (Fox-Kemper et al., 2021a), and an enhanced ESL module could incorporate parametric representations of such changes. A further refinement could capture the relationship between these changes and GSAT, allowing the analysis to reflect correlations between RSL change and storm changes (Lockwood et al., 2022). Alternative ESL modules might also use different data sources, such as regional or global hydrodynamic models (e.g., Dullaart et al., 2020).

Contemporary GRD processes in current FACTS modules are currently based on a library of scaling factors (sometimes called 'fingerprints') applied to ice sheet, regional glacier, and land water storage projections in each module's post-processing Stage. Such a library approach is most appropriate for glaciers, as the glacier regions are geographically small enough that the shifts in the locus of mass loss within a region will not substantially modify that region's fingerprint. For the larger ice sheets, however, the locus of ice mass change can significantly affect the contemporary GRD spatial pattern (Larour et al., 2017; Mitrovica et al., 2018; Cederberg et al., 2023). This variability could be incorporated into FACTS through more spatially resolved ice sheet emulation, as well as potentially through a new integration module that includes an online GRD solver (e.g., Larour et al., 2020).

The existing FACTS modules start projections in the 21st century. This choice is, in part, a limitation of the underlying studies on which these modules are built. While CMIP6 historical climate simulations start in 1850, neither the GlacierMIP nor ISMIP6 model intercomparison exercises include historical simulations (Nowicki et al., 2016, 2020; Hock et al., 2019; Marzeion et al., 2020). Implementing work-arounds to these absences – and incorporating new historical simulations as they become available – would allow FACTS projections to start in the 19th or 20th century, and thus enable model/data comparison for changes in both sea level and individual components. This, in turn, could allow the probability distributions generated by FACTS to be updated in a Bayesian sense based on historical, current, and (as time proceeds) future observations. While current observations are unlikely to significantly reduce the deep uncertainty associated with late-21st century high-end projections (Kopp et al., 2017), they could have a substantial impact on nearer-term projections. A Bayesian approach could also be coupled to economic models (e.g. ?) to assess the value of information associated with additional observational constraints and process-model enhancements.

Relevant to such a model-data comparison is that the existing FACTS sea-level component modules are focused on projecting changes in tidal-datum-epoch (i.e., 19-year) average mean sea level, not higher temporal frequency (e.g., interannual) variabil-

ity. Alternative ocean dynamics modules could introduce this higher frequency variability, or, alternatively, the auto-correlation structure of such variability could be incorporated into model/data comparisons.

Some modeling approaches may require more communication between modules. At the moment, all sea-level components are computed independently, conditional upon a common input of projected GSAT and/or ocean heat content in the climate step. As a consequence, uncertainty within individual Workflows is probably underestimated (e.g., Le Bars, 2018; van de Wal

et al., 2022) While correlations between GSAT and global ocean heat content change will tend to lead to some correlation between projected sea-level components (e.g., Palmer et al., 2020), correlations associated with regional or systematic changes are not represented. This is not in the case in Earth system models, where (for example) meltwater input affects sterodynamic sea level (e.g., Lambert et al., 2021). Representations of such interactions could be incorporated into FACTS by subdividing the sea-level components Experiment Steps, either recursively refining projections with one-way coupling (e.g., modifying an

initial dynamic sea-level projection for meltwater input) or proceeding in incremental time steps with two-way coupling.

More broadly, to date, FACTS has been developed by a small team, with a primary objective being to support specific assessment processes, particularly that of the IPCC AR6. A critical objective moving forward is to transform FACTS into a larger-scale community project, with modules developed autonomously by different research and assessment teams. The structure of FACTS – which enables modules to serve as wrappers around independently developed code – is intended to

facilitate such efforts.

## 5  Conclusions

Sea-level rise is a major driver of climate risk to coastal communities and ecosystems around the world. Appropriately managing this risk requires planners to be cognizant of both quantifiable and structural uncertainty in projections of future sea-level change, and synthesizing this information is an important task of scientific assessment processes. FACTS provides a flexible,

modular, and open-source platform that allows comparable probabilistic outputs to be generated in parallel through multiple modeling approaches. Its flexibility allows it to be customized based on the needs of specific assessment processes (e.g., substituting alternative approaches to VLM or higher-resolution sterodynamic sea level), while its parallel Workflow structure supports the characterization of deep uncertainty.

*Code availability.*  The development version of FACTS is available under a MIT license in a Git-version controlled repository at https:

//github.com/radical-collaboration/facts. The latest release is archived on Zenodo at https://doi.org/10.5281/zenodo.7502824. Documentation is included in the repository.

*Data availability.*  Input data sets for the modules described in this manuscript are available on Zenodo at https://doi.org/10.5281/zenodo.7478191 and https://doi.org/10.5281/zenodo.7478447. Summary data sets describing the IPCC AR6 sea-level projections are available on Zenodo at https://doi.org/10.5281/zenodo.5914709.

# A `tlm/sterodynamics` methodology

The ocean dynamic sea-level projection method used by the `tlm/sterodynamics` module is a modification of that described in Kopp et al. (2014). Whereas in Kopp et al. (2014) global-mean thermosteric sea-level rise projections are derived directly from a GCM ensemble, in `tlm/sterodynamics` they are generated from the two-layer model, as described in Fox-Kemper et al. (2021b).

As in Kopp et al. (2014), ocean dynamic sea level is assumed to have a degree of correlation with global-mean thermosteric sea-level rise, with the correlation assessed on a grid-cell basis. In the case of `tlm/sterodynamics`, the correlation is calculated based on the CMIP6 ensemble for a particular (specified) SSP scenario. Given a sample of 19-year-average global-mean thermosteric sea-level rise $y$ at a particular point in time $t$, 19-year-average ocean dynamic sea level $z$ is taken as distributed following a t-distribution with a conditional mean of

$$\bar{z}_t(r) + \sigma_t(r)k_t(r)\frac{y_t - \bar{y}_t}{s_t} \tag{A1}$$

and a conditional standard deviation proportional to

$$\sigma_t(r)1 - k_t(r)^2, \tag{A2}$$

Where $z_t(r)$ is the multimodel mean ocean dynamic sea level at time $t$ and location $r$, $\sigma_t(r)$ is the multimodel standard deviation, $k_t(r)$ is the correlation between global-mean thermosteric sea-level rise and $z_t(r)$, $\bar{y}_t$ is the multi-model mean of global-mean thermosteric sea-level rise, and $s_t$ is the standard deviation across models of global-mean thermosteric sea-level rise. The standard deviation is inflated relative to that of the ensemble to account for the expert judgment that the 5-95th percentile of the ensemble may have as much as a 33% of being exceeded on either end (ie the 5-95th percentile range is treated as a likely range). Though the parameters of this regression model are re-fit for each time point, correlation across time is preserved (perhaps excessively) in sampling by drawing (via Latin hypercube sampling) a single quantile of the variance characterized by the conditional standard deviation to use at all time points for a given time series sample. In sampling the t-distribution, the number of degrees of freedom is taken as the number of GCMs providing DSL projections for a particular grid cell in the scenario used for calibration.

In some ways, the approach is similar to that of a linear-regression based scaling of ocean dynamic sea level on global-mean thermosteric sea-level rise, as in Palmer et al. (2020). The commonality is the assumption that the distribution of ocean dynamic sea level at a given point may be constrained by information about global-mean thermosteric sea-level rise. ("May" is an operative word here — it is also possible for the scaling factor or correlation coefficient to be zero).

One important difference is that this approach is recalibrated for each time step, whereas the Palmer et al. (2020) approach finds a single regression coefficient for a given GCM across time. A second is that the uncertainty not captured in the characterized correlation is sampled, whereas in Palmer et al. (2020), all variance is assumed to be captured by the spread of regression coefficients across GCMs. The approach used here is more focused on the distributional characteristics across GCMs, as opposed to representing each individual GCM by a regression coefficient. As a consequence of these differences, the Kopp et al.

**Table A1.** GMSL Component Projections for 2100 Including AR6 Projections

| Component | Label | SSP1-2.6 | SSP2-4.5 | SSP5-8.5 |
|---|---|---|---|---|
| Glaciers | `emulandice/glaciers` | 0.09 (0.07–0.11) | 0.12 (0.10–0.14) | 0.18 (0.15–0.20) |
| Glaciers | AR6 emulated GlacierMIP (Table 9.4) | 0.08 (0.06–0.10) | 0.12 (0.09–0.14) | 0.17 (0.14–0.20) |
| Glaciers | `ipccar5/glaciers` (GMIP2) | 0.09 (0.06–0.13) | 0.12 (0.08–0.16) | 0.16 (0.11–0.22) |
| Glaciers | GlacierMIP parametric fit (Table 9.4) | 0.10 (0.08–0.13) | 0.13 (0.10–0.17) | 0.17 (0.12–0.22) |
| Antarctica | `bamber19/icesheets` | 0.10 (-0.01–0.26) | — | 0.20 (0.02–0.57) |
| Antarctica | AR6 SEJ (Table 9.8) | 0.09 (-0.01–0.25) | — | 0.21 (0.02–0.56) |
| Antarctica | `deconto21/AIS` | 0.08 (0.06–0.11) | 0.09 (0.07–0.11) | 0.34 (0.19–0.53) |
| Antarctica | AR6 MICI (Table 9.3) | 0.08 (0.06–0.12) | 0.09 (0.07–0.11) | 0.34 (0.19–0.53) |
| Antarctica | `emulandice/AIS` | 0.08 (0.03–0.14) | 0.08 (0.03–0.14) | 0.08 (0.03–0.14) |
| Antarctica | AR6 emulated ISMIP6 (Table 9.3) | 0.09 (0.03–0.14) | 0.09 (0.03–0.14) | 0.08 (0.03–0.14) |
| Antarctica | `larmip/AIS` | 0.13 (0.05–0.26) | 0.14 (0.05–0.29) | 0.15 (0.05–0.34) |
| Antarctica | AR6 LARMIP-2 with SMB (Table 9.3) | 0.13 (0.06–0.27) | 0.14 (0.06–0.29) | 0.15 (0.05–0.34) |
| Greenland | `bamber19/icesheets` | 0.13 (0.07–0.30) | — | 0.22 (0.10–0.59) |
| Greenland | AR6 SEJ (Table 9.8) | 0.13 (0.07–0.30) | — | 0.23 (0.10–0.59) |
| Greenland | `emulandice/GrIS` | 0.05 (0.01–0.10) | 0.08 (0.04–0.13) | 0.12 (0.08–0.18) |
| Greenland | AR6 emulated ISMIP6 (Table 9.2) | 0.06 (0.01–0.10) | 0.08 (0.04–0.13) | 0.13 (0.09–0.18) |
| Greenland | `FittedISMIP/GrIS` | 0.08 (0.06–0.10) | 0.10 (0.08–0.12) | 0.14 (0.11–0.18) |
| Greenland | AR6 parametric ISMIP fit (Table 9.2) | 0.08 (0.06–0.10) | 0.10 (0.08–0.13) | 0.14 (0.11–0.18) |
| Land Water Storage | `ssp/landwaterstorage` | 0.03 (0.02–0.04) | 0.03 (0.02–0.04) | 0.03 (0.02–0.04) |
| Land Water Storage | AR6 land-water storage (Table 9.9) | 0.03 (0.01–0.04) | 0.03 (0.01–0.04) | 0.03 (0.01–0.04) |
| Thermal Expansion | `tlm/sterodynamics` | 0.14 (0.11–0.17) | 0.19 (0.15–0.23) | 0.29 (0.24–0.35) |
| Thermal Expansion | AR6 thermal expansion (Table 9.9) | 0.14 (0.11–0.18) | 0.20 (0.16–0.24) | 0.30 (0.24–0.36) |

Median (17th-83rd percentile) projections are shown relative to a 1995–2014 baseline. All are in meters. For certain modules, projections for Representative Concentration Pathways 2.6, 4.5, and 8.5 are shown in lieu of those for the SSP scenarios. AR6 results taken from Fox-Kemper et al. (2021a) Tables 9.2, 9.3, 9.4, 9.8, and 9.9, as indicated by numbers in parentheses after label.

(2014) approach loses a degree of traceability to individual GCMs, being instead focused on preserving the distributional properties assessed based on the ensemble.

Note that where global-mean thermosteric sea-level rise and ocean dynamic sea level are uncorrelated, this approach returns simply the multimodel mean and scaled standard deviation for the scenario.

**Table A2.** Total Projections for 2100 compared to AR6

| Workflow | SSP1-1.9 | SSP1-2.6 | SSP2-4.5 | SSP3-7.0 | SSP5-8.5 |
|---|---|---|---|---|---|
| **Global-mean sea level – FACTS 1.0 workflows** | | | | | |
| 1e | 0.35 (0.27–0.44) | 0.40 (0.32–0.49) | 0.50 (0.42–0.60) | 0.62 (0.53–0.73) | 0.71 (0.61–0.82) |
| 1f | 0.35 (0.27–0.44) | 0.40 (0.31–0.49) | 0.49 (0.40–0.59) | 0.58 (0.48–0.68) | 0.66 (0.55–0.78) |
| 2e | 0.40 (0.30–0.53) | 0.46 (0.35–0.60) | 0.57 (0.45–0.73) | 0.70 (0.57–0.88) | 0.80 (0.65–1.00) |
| 2f | 0.41 (0.33–0.54) | 0.48 (0.38–0.62) | 0.59 (0.48–0.74) | 0.70 (0.58–0.87) | 0.80 (0.66–1.00) |
| 3e | — | 0.40 (0.34–0.48) | 0.51 (0.45–0.59) | — | 0.97 (0.80–1.18) |
| 3f | — | 0.43 (0.37–0.49) | 0.53 (0.47–0.61) | — | 0.97 (0.81–1.17) |
| 4 | — | 0.53 (0.37–0.80) | — | — | 1.01 (0.69–1.64) |
| **Global-mean sea level – AR6 p-boxes** | | | | | |
| *Medium confidence* | 0.38 (0.28–0.55) | 0.44 (0.32–0.62) | 0.56 (0.44–0.76) | 0.68 (0.55–0.90) | 0.77 (0.63–1.01) |
| *Low confidence* | — | 0.45 (0.32–0.79) | — | — | 0.88 (0.63–1.60) |

Median (17th-83rd percentile) projections are shown in meters relative to a 1995-2014 baseline. AR6 values taken from Fox-Kemper et al. (2021a) Table 9.9, except for *low confidence* SSP1-2.6 values, taken from Garner et al. (2021). Table 9.9 results are based on workflows 1e and 2e (*medium confidence* projections) and workflows 1e, 2e, 3e, and 4 (*low confidence* projections).

*Author contributions.* REK, ABAS, and SJ conceived the project, and REK supervised and administered the project. GGG developed the FACTS architecture and most of the FACTS modules. REK, PK, AR, TH, MT, AM, GK, and SJ contributed to code development. TLE (emulandice), THJH (extremesealevel), REK (kopp14), AL (larmip), SN (emulandice), JG (ipccar5), MDP (fair) and CS (fair) led the initial development of individual module sets. All authors contributed to the writing and editing of the manuscript.

*Competing interests.* The authors declare that they have no conflict of interest.

*Acknowledgements.* GGG and REK were supported by grants from the National Science Foundation (ICER-1663807 and ICER-2103754, the latter as part of the Megalopolitan Coastal Transformation Hub) and the National Aeronautics and Space Administration (grants 80NSSC17K0698, 80NSSC20K1724 and 80NSSC21K0322, and JPL task 105393.509496.02.08.13.31). REK was also supported by a grant from the Rhodium Group (for whom he has previously worked as a consultant) as part of the Climate Impact Lab consortium. This project was supported by the 610 U.K. Natural Environment Research Council grant NE/T007443/1. ABAS and THJH were supported by NIOZ Royal Netherlands Institute for Sea Research. ABAS, THJH and TLE were supported by PROTECT, which has received funding from the European Union's Horizon 2020 research and innovation programme under grant agreement no. 869304. (This is PROTECT contribution number 78). This project was also supported by the NZ SeaRise Programme funded by New Zealand Ministry of Business, Innovation & Employment Contract to the Research Trust at Victoria University Contract ID - RTVU1705. MDP was supported by the Met Office Hadley Centre Climate Programme 615 funded by BEIS. CS was supported by a NERC/IIASA collaborative research fellowship (NE/T009381/1).

**Table A3.** CMIP6 models used for calibrating the thermal expansion coefficients of Fox-Kemper et al. (2021a) (TE, left column) and for projecting ocean dynamic sea-level change and the IB effect (zos+psl, right column) in the `tlm/sterodynamics` module

| Model | TE | zos+psl |
|---|---|---|
| ACCESS-CM2 | x | x |
| ACCESS-ESM1-5 | x | x |
| BCC-CSM2-MR | | x |
| BCC-ESM1 | | x |
| CAMS-CSM1-0 | | x |
| CanESM5 | x | x |
| CanESM5-CanOE | | x |
| CAS-ESM2-0 | | x |
| CESM2 | | x |
| CESM2-FV2 | | x |
| CESM2-WACCM | | x |
| CESM2-WACCM-FV2 | | x |
| CIESM | | x |
| CMCC-CM2-SR5 | | x |
| CNRM-CM6-1 | x | x |
| CNRM-CM6-1-HR | x | x |
| CNRM-ESM2-1 | x | x |
| EC-Earth3 | x | x |
| EC-Earth3-Veg | x | x |
| EC-Earth3-Veg-LR | | x |
| FIO-ESM-2-0 | | x |
| GISS-E2-1-G | | x |
| GISS-E2-1-G-CC | | x |
| HadGEM3-GC31-LL | x | x |
| HadGEM3-GC31-MM | | x |
| INM-CM4-8 | | x |
| INM-CM5-0 | x | x |
| IPSL-CM6A-LR | x | x |
| MIROC6 | x | x |
| MIROC-ES2L | | x |
| MPI-ESM-1-2-HAM | | x |
| MPI-ESM1-2-HR | x | x |
| MPI-ESM1-2-LR | x | x |
| MRI-ESM2-0 | x | x |
| NorCPM1 | | x |
| NorESM2-LM | x | x |
| NorESM2-MM | x | x |
| UKESM1-0-LL | x | |

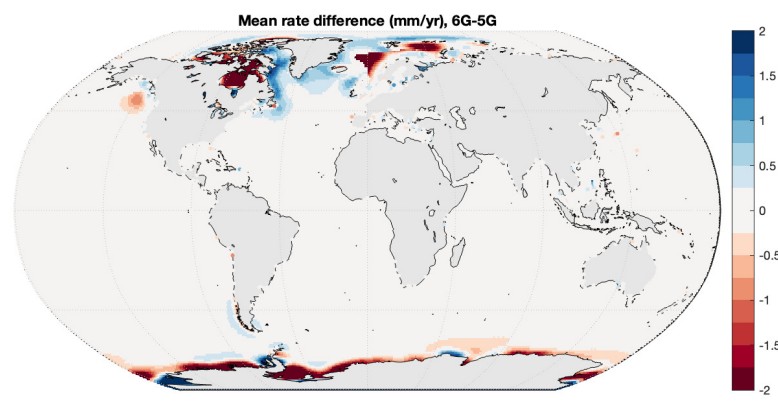

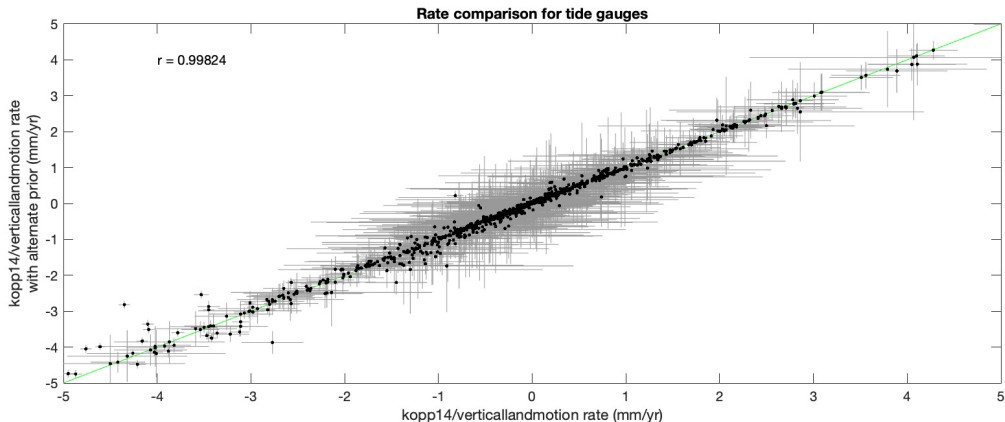

**Figure A1.** Comparison of mean rate estimates from the `kopp14/verticallandmotion` module, which uses the ICE5G ice history and VM2-90 viscosity profile (Peltier, 2004), with rate estimates derived using the same methodology but with a prior based on the ICE-6G-C ice history and VM5a viscosity profile (Stuhne and Peltier, 2015). (a) Differences in absolute rates (mm/yr) at tide gauges and grid cells. Pale areas have an absolute mean rate difference of $< 0.25$ mm/yr. (b) For tide gauge locations, rate estimate derived using the alternative prior with the `kopp14/verticallandmotion` estimate. Green line is 1:1. Uncertainties shown are $\pm 1\sigma$.

We acknowledge the World Climate Research Programme, which, through its Working Group on Coupled Modelling, coordinated and promoted CMIP6. We thank the climate modeling groups for producing and making available their model output, the Earth System Grid Federation (ESGF) for archiving the data and providing access, and the multiple funding agencies who support CMIP6 and ESGF.

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
