# Peer review of "The Framework for Assessing Changes To Sea-level (FACTS) v1.0: A platform for characterizing parametric and structural uncertainty in future global, relative, and extreme sea-level change"

_EGUsphere, 2023_

## Author Response (AR1)

**Benjamin Grandey [03 Feb 2023]**

> The authors provide a clear description and helpful discussion of a useful tool, FACTS.

> I have a few minor comments and questions for the authors' consideration.

> L53: The acronym FAIR is used in two senses: (i) "Findable, Accessible, Interoperable, Reusuable" (only once, at L53), and (ii) the FAIR climate model. Only the first of these is defined. To avoid potential confusion, could the acronym be reserved for the FAIR climate model?

Thank you for the suggestion. We will replace the reference to "FAIR science" to "open science". We have also adjusted the acronym used for the FaIR climate model to have a lowercase 'a', which has become the preferred usage.

> L165: Should "FAIR 1.0" not be "FACTS 1.0"?

You are correct. Thank you for correcting the error.

> L250-252: When calculating thermosteric sea level rise, the authors state that the tlm/sterodynamics module uses ocean heat content (OHC) from the fair/temperature module, following Fox-Kemper et al. (2021). Does FAIR produce the OHC data, or is the climate simulation step bypassed (L150)? I understand that Fox-Kemper et al. (2021) used OHC from a two-layer energy balance model. Have I misunderstood something?

The two-layer energy balance model is incorporated into FaIR as an alternative representation of the forcing-temperature coupling, based on Geoffroy et al. (2013). (See https://docs.fairmodel.net/en/v1.6.3/examples.html#geoffroy-temperature-function) . We will add a parenthetical comment in the text to this effect.

> Table 1: Does every module sample a distribution? Or do some modules produce only a single time series? In particular, I am a little confused by the nature of the output from the land water storage and vertical land motion modules (L271-297).

Yes, every module produces a distribution. For land water storage, we will note: "Uncertainty in the projections is generated by sampling the parameters of the sigmoidal fit for reservoir storage and linear fit for groundwater depletion." For VLM: "Uncertainty in the projections is generated based on the uncertainty in the estimate of the constant trend."

> L345: Should "Table 2" not be "Table 1"?

Yes, thank you.

L375-378: I understand that Bamber et al. (2019) sought to account for dependence between the ice sheet components. Is this dependence preserved in workflow 4? Does this explain the large positive interaction evident in the early decades of workflow 4 (Fig. 4 bottom row)? If so, why does this positive interaction decrease in later decades?

The samples used in workflow 4 for the ice sheets are directly taken from those reported by Bamber et al. (2019), preserving all correlations therein. While a deep dive into the correlations is beyond the scope of this manuscript, it is by construction consistent with the Monte Carlo samples of ice sheets generated in Bamber et al 2019, because the samples are used directly as a static input. Bear in mind that (1) the absolute variance early in the 21st century are quite small, and (2) correlations in Bamber et al. (2019) were not elicited early in the 21st century, so I would read too much into the high positive correlation in the early years of the decade – the import of this correlation is small, given the small overall variance.

Figures 2, 3, and 5 captions: Should "think" not be "thin"?

Yes, thank you.

**Dewi Le Bars [06 Feb 2023]**

Dear authors,

Thank you for this effort to clearly describe this framework and to make the code openly available. This work is of great value to the community.

I do not understand the description of the method for ODSL in l.255-258:

"The resulting global mean thermosteric sea-level rise is then combined with ocean dynamic sea-level change and the inverse barometer effect using the gridded output of CMIP6 models (see the right column of Table A1 for the models that were used in (Fox-Kemper et al., 2021)), based on the time-varying correlation structure between global mean thermal expansion and ocean dynamic sea-level change in the multi-model ensemble."

I don't think that a correlations structure is enough information to reconstruct ODSL from global mean thermal expansion. Is this method similar to Palmer et al. 2020? (https://agupubs.onlinelibrary.wiley.com/doi/full/10.1029/2019EF001413) There, it is described as a linear regression but instead of being between global mean thermal expansion and ODSL it is between global mean thermal expansion and sterodynamic sea-level change:

"Following previous studies (Bilbao et al., 2015; Palmer, Howard, et al., 2018; Perrette et al., 2013), the effects of local changes in ocean density and circulation are included by establishing regression relationships between global thermal expansion and local sterodynamic sea-level change in CMIP5 climate model simulations"

Your clarifications on this point would be very much appreciated. In some places the assumption of linear regression between global mean thermal expansion and ocean dynamic sea-level change is not very accurate, therefore making this assumption clearer would help people decide if this framework works for them or not.

Thank you,

Dewi Le Bars

Dear Dewi,

Thank you for the question. We will add an appendix to the following effect:

The method used is a modification of that described in Kopp et al. (2014). Global mean thermal expansion projections are generated from the two-layer model. Ocean dynamic sea level is assumed to have a degree of correlation with global mean thermal expansion, with the correlation assessed on a grid-cell basis based on the CMIP6 ensemble for a particular SSP scenario. Given a sample of 19-year-average global mean thermal expansion $y$ at a particular point in time, 19-year-average ocean dynamic sea level is taken as distributed following a $t$ distribution with a conditional mean of

$$\bar{z}(r) \; + \; \sigma(r)\, k(r)\, (y \, - \, \bar{y}\,)/s$$

And a conditional standard deviation proportional to

$$\sigma(r)\, \sqrt{1 - k(r)^2}$$

Where $\bar{z}(r)$ is the multimodel mean ocean dynamic sea level at location $r$, $\sigma(r)$ is the multimodel standard deviation, $k(r)$ is the correlation between global mean thermal expansion and ocean dynamic sea level $z(r)$, $\bar{y}$ is the multimodal mean global mean thermal expansion and $s$ is the multimodel standard deviation of global mean thermal expansion. The standard deviation is inflated relative to that of the ensemble to account for the expert judgment that the 5-95th percentile of the ensemble may have as much as a 33% of being exceeded on either end (ie the 5-95th percentile range is treated as a *likely* range). Though the parameters of this regression model are re-fit for each time point, correlation across time is preserved (perhaps excessively) in sampling by drawing (via Latin hypercube sampling) a single quantile of the variance characterized by the conditional standard deviation to use at all time points for a given time series sample. In sampling the t-distribution, the number of degrees of freedom is taken as the number of GCMs providing DSL projections for a particular grid cell in the scenario used for calibration.

In some ways, the approach is similar to that of a linear-regression based scaling of ocean dynamic sea level on global mean thermal expansion, as in Palmer et al. (2020). The commonality is the assumption that the distribution of ocean dynamic sea level at a given point may be constrained by information about global mean thermal expansion. ("May" is an operative word here — it is also possible for the scaling factor or correlation coefficient to be zero).

One important difference is that this approach is recalibrated for each time step, whereas the Palmer et al. approach finds a single regression coefficient for a given GCM across time. A second is the uncertainty not captured in the characterized correlation is sampled, whereas in Palmer et al., all variance is assumed to be captured by the spread of regression coefficients across GCMs. The approach used here is more focused on the distributional characteristics across GCMs, as opposed to representing each individual

GCM by a regression coefficient. As a consequence of these differences, the Kopp et al. (2014) approach loses a degree of traceability to individual GCMs, being instead focused on preserving the distributional properties assessed based on the ensemble.

Note that where thermal expansion and ocean dynamic sea level are uncorrelated, this approach returns simply the multimodel mean and scaled standard deviation for the scenario.

Best,

The authors

**Vanessa Volz [08 Feb 2023]**

Very helpful discription of FACTS!

Chapter 3, L334: Aren't there 20,000 instead of 2,000 Monte Carlo samples?

Thank you! This manuscript describes FACTS 1.0 and demonstrates it using a set of modules that were developed, in part, to support the AR6 assessment. It does not document the AR6 sea level projections, for which 20,000 Monte Carlo samples were run; the demonstration here uses 2,000 Monte Carlo samples.

**Referee #1 [25 Apr 2023]**

> The authors present the FACTS framework to probabilistically estimate future sea level rise, globally and regionally. The framework aims to make it possible to seamlessly exchange individual drivers of global sea level rise so structural uncertainty can be explored. The presented work stands out as it underpins several authoritative sea level assessments, of which the most prominent is the IPCC AR6 WG1 assessment.
>
> Due to this special position of FACTS, usability and replicability is a key concern. I therefore split this review into two parts: part one is on the scientific aspects and clarity. My comments here are mainly on clarification and better explanation because the method is a continuation of established works and because the AR6 methodology is fixed and I see a major function of the manuscript to document that methodology. Part two is on usability and replicability: readers should be able to replicate AR6 sea level numbers with the manuscript and the code at hand without being experts in specific high performance computing environments. I tried but failed. (I followed the rather succinct "Quick Start" documentation.) I propose improvements to be made to the manuscript and to the code to overcome this. Only if I (as an example user) succeed in a "replication of the AR6 approach entirely within FACTS " (stated in line 72-73) the work can reach its full potential and follow its aspiration to become a "larger-scale community project" (line 505).

We thank the reviewer for their comments.

We should clarify that the goal of the paper is to document FACTS 1.0, not document the production of the AR6 sea level projections. While much of what we describe provides useful detail for interpreting how AR6 produced its projections, FACTS development began well before it became the preferred tool for AR6, and continued after the completion of AR6. A number of modeling steps that were done via offline coupling (e.g., WG1 chapter 7 produced FaIR projections, which were handed off to the chapter 9 emulandice and LARMIP emulators to produce land ice time series, which were then taken as input to FACTS modules alongside the temperature and ocean heat content trajectories) are now done through coupling within the FACTS framework. Minor numerical differences are unsurprising, but the output for individual modules all agree with the results presented in AR6 within 0.01 m rounding errors, as we present in the revision:

**Table A1.** GMSL Component Projections for 2100 compared to AR6

| Component | Label | SSP1-2.6 | SSP2-4.5 | SSP5-8.5 |
|---|---|---|---|---|
| Glaciers | `emulandice/glaciers` | 0.09 (0.07–0.11) | 0.12 (0.10–0.14) | 0.18 (0.15–0.20) |
| Glaciers | AR6 emulated (9.4) | 0.08 (0.06–0.10) | 0.12 (0.09–0.14) | 0.17 (0.14–0.20) |
| Glaciers | `ipccar5/glaciers` (GMIP2) | 0.09 (0.06–0.13) | 0.12 (0.08–0.16) | 0.16 (0.11–0.22) |
| Glaciers | GlacierMIP parametric fit (9.4) | 0.10 (0.08–0.13) | 0.13 (0.10–0.17) | 0.17 (0.12–0.22) |
| Antarctica | `bamber19/icesheets` | 0.10 (-0.01–0.26) | — | 0.20 (0.02–0.57) |
| Antarctica | AR6 SEJ (9.8) | 0.09 (-0.01–0.25) | — | 0.21 (0.02–0.56) |
| Antarctica | `deconto21/AIS` | 0.08 (0.06–0.11) | 0.09 (0.07–0.11) | 0.34 (0.19–0.53) |
| Antarctica | AR6 MICI (9.3) | 0.08 (0.06–0.12) | 0.09 (0.07–0.11) | 0.34 (0.19–0.53) |
| Antarctica | `emulandice/AIS` | 0.08 (0.03–0.14) | 0.08 (0.03–0.14) | 0.08 (0.03–0.14) |
| Antarctica | AR6 emulated ISMIP6 (9.3) | 0.09 (0.03–0.14) | 0.09 (0.03–0.14) | 0.08 (0.03–0.14) |
| Antarctica | `larmip/AIS` | 0.13 (0.05–0.26) | 0.14 (0.05–0.29) | 0.15 (0.05–0.34) |
| Antarctica | AR6 LARMIP-2 with SMB (9.3) | 0.13 (0.06–0.27) | 0.14 (0.06–0.29) | 0.15 (0.05–0.34) |
| Greenland | `bamber19/icesheets` | 0.13 (0.07–0.30) | — | 0.22 (0.10–0.59) |
| Greenland | AR6 SEJ (9.8) | 0.13 (0.07–0.30) | — | 0.23 (0.10–0.59) |
| Greenland | `emulandice/GrIS` | 0.05 (0.01–0.10) | 0.08 (0.04–0.13) | 0.12 (0.08–0.18) |
| Greenland | AR6 emulated ISMIP6 (9.2) | 0.06 (0.01–0.10) | 0.08 (0.04–0.13) | 0.13 (0.09–0.18) |
| Greenland | `FittedISMIP/GrIS` | 0.08 (0.06–0.10) | 0.10 (0.08–0.12) | 0.14 (0.11–0.18) |
| Greenland | AR6 parametric ISMIP fit (9.2) | 0.08 (0.06–0.10) | 0.10 (0.08–0.13) | 0.14 (0.11–0.18) |
| Land Water Storage | `ssp/landwaterstorage` | 0.03 (0.02–0.04) | 0.03 (0.02–0.04) | 0.03 (0.02–0.04) |
| Land Water Storage | AR6 land-water storage (9.9) | 0.03 (0.01–0.04) | 0.03 (0.01–0.04) | 0.03 (0.01–0.04) |
| Thermal Expansion | `tlm/sterodynamics` | 0.14 (0.11–0.17) | 0.19 (0.15–0.23) | 0.29 (0.24–0.35) |
| Thermal Expansion | AR6 thermal expansion (9.9) | 0.14 (0.11–0.18) | 0.20 (0.16–0.24) | 0.30 (0.24–0.36) |

Median (17th-83rd percentile) projections are shown relative to a 1995–2014 baseline. All are in m except for global mean surface air temperature (GSAT), which is in °C above a 1850–1900 baseline. For certain modules, projections for Representative Concentration Pathways 2.6, 4.5, and 8.5 are shown in lieu of those for the SSP scenarios. AR6 results taken from Tables 9.2, 9.3, 9.4, 9.8, and 9.9, as indicated by numbers in parentheses after label.

In addition, the AR6 presentation of sea-level rise projections is in an imprecise probability mode, described in AR6 WG1 9.6.3.2, in which results are summarized by p-boxes that encompass multiple alternative probability distributions. FACTS' job is to produce samples from multiple alternative probability distributions of sea level rise, not to prescribe a particular approach to how these distributions are summarized and combined. The particular historical, epistemological, and communicative rationale underlying the AR6 approaches are described in a separate manuscript (Kopp et al., 2023, https://doi.org/10.1038/s41558-023-01691-8), and are not the subject of this manuscript. We believe some of the reviewer's questions about discrepancies between the numerical results in this paper and those in the manuscript relate to the difference between individual probability distributions (shown here) and p-boxes (as shown in WG1 Table 9.9, and other figures and tables in the chapter).

The distinction is illustrated by Fig. 1 in Kopp et al. (2023) referenced above, and reproduced here:

[Figure]

The first two rows illustrate individual probability distributions (corresponding to WF 1e, 2e, 3e, and 4 in the current manuscript) as (a,b) probability distribution functions and (c,d) cumulative distribution functions. The last row (e,f) represents how these distributions are summarized in AR6 by *medium confidence* and *low confidence* p-boxes. The results presented in the current manuscript and produced by FACTS correspond to the distributions shown in the top two rows; the results presented in AR6 correspond to the bottom row.

FACTS per se does not produce the final row, and cannot, since its basic mode of operation is to sample probability distributions corresponding to different workflows. Synthesizing these results in a summary form is necessarily a post-processing step. We are adding Jupyter notebooks to the Github repo to facilitate this step, but this is not a core part of FACTS proper.

We also confused the reviewer by using the term 'vertical land motion' to refer to long-term vertical land motion, of the sort reflected in century-scale analysis of tide-gauge data. Elastic deformation associated with contemporary land-ice and land-water mass redistribution is accounted for in the RSL projections via the static GRD fingerprints. This is now clarified in the manuscript.

> Part 1:
>
> I have four points that need more clarity in my view.
>
> 1) It is not straightforward to understand how the IPCC AR6 numbers are derived from FACTS. It is described in L411ff, which is already part of the discussion section. The manuscript would profit to state upfront how the AR6 numbers are constructed within the manuscript, for example as part of sec 3.3 or as a separate section. I also advocate for stating the AR6 numbers directly within the manuscript (i.e. within tables 3 and 4), which would make comparison easier. For now I find close correspondence, but no replication of IPCC AR6 numbers (from WG1 Table 9.9). For replication I would expect the numbers to match. If not, I would at least expect a paragraph where the numbers are related and differences justified. Ideally the setup for AR6 replication (a "cookbook") would be prepared within the codebase so that the user does not have to manually infer the settings from the manuscript.

As noted above, we added a table showing that the module output presented in the paper agrees with the AR6 table results within 0.01 m rounding errors. More substantial differences identified by the reviewer may be partially the result of confusing summary p-box distributions with the output of individual workflows; we have attempted to clarify this point by indicating which workflows are used for which p-box in Table 2.

> 2) VLM is now recognized as a key driver of relative sea level rise and thus impacts (i.e. Nicholls et al, 2021), but it does not get the necessary attention in the manuscript.
>
> a) VLM estimation is based on Kopp et al. 2014, which uses a Gaussian process model to fit historical tide gauge data. The step from fitting to tide gauge data (yielding spatial fields of relative sea level as output) to estimating the VLM component is not clear to me even after reading the SI of Kopp et al. 2014. This needs additional explanation.

The model approximates VLM (and the geocentric sea level contribution of GIA) as the century-scale constant rate trend that remains after removing the global-mean signal, the regionally and temporally-correlated non-linear signal. The prior estimate of this trend is taken from a GIA model (ICE5G VM2-90), to which the estimate of the trend reverts (with added uncertainty) as one makes predictions at greater distances from the data.

b) To my knowledge the approach does not involve  direct observations of VLM (i.e. GNSS), where much progress has been seen for VLM estimation. For example, involving such measurements to correct tide gauge measurements for VLM crucially helped to close the sea level budget (Frederikse et al 2020). Can you justify the default choice of Kopp 2014?

The observational record of VLM remains relatively short in most of the world; the majority of GNSS stations at tide gauges were installed in the 21st century . The long-term VLM module assumes rates of VLM can be extrapolated forward at a constant rate through 2150. The tide-gauge based estimates, which average over a longer period, are better suited for this future than the available GNSS data in many regions. In addition, note that this module is intended to represent GIA in full, not just the GIA contribution to vertical land motion. The sea-surface height effect of GIA is incorporated into the estimated long-term trend based on tide-gauge analysis per Kopp et al., 2014, but would be neglected if only GNSS results were used.

In general, changes to VLM based on local knowledge are the most requested revision to the AR6 projections by stakeholders. This would likely be the case for any global analysis method.

c) Changes in contemporary ice mass loading affect not only the ocean water distribution (which I see included through the GRD fingerprints), but also VLM. GRD fingerprints are mentioned for ice sheets (l222), but the reference is more than 20 years old and it is not clear if newer works, especially these of Thomas Frederikse (2017, 2019, 2020) are represented and how and if they affect the VLM estimates of the presented work.

Elastic deformation (the 'D' of GRD) is included in the GRD fingerprints used in localizing the land-ice and land-water storage results. Mitrovica et al. (2001) is cited as a seminal paper that first incorporated the 'R' in GRD. (The 'G' and 'D' were modeled by Clark and Lingle, 1977). We have added a reference to Gomez et al. (2010), which provides a better historical presentation of the development of GRD theory, as well as detailed theoretical derivation. The understanding of elastic GRD physics has not changed substantially since then. (Our projections do not include viscoelastic effects, which recent work as shown on a century + timescale might be significant in low-viscosity regions like West Antarctica; e.g., Pan et al., 2021) The Frederikse work cited deals with the detection of fingerprints in GRACE data and the interpretation of sea-level observations, not GRD projections.

d) VLM is independent of future warming, which could be said clearer (currently

referred to as "constant trend", l285) and also stated as a caveat: future ice mass loss is scenario dependent and will influence the VLM rate but it is not implemented in FACTS.

The deformation associated with future, scenario-dependent ice-mass loss (and changes in land-water storage) is incorporated into the GRD fingerprints applied to these components.

3) It is not clear to which baseline the individual contributions are referenced to. Though the authors mention Gregory et al. 2019, which did an excellent job on clarifying terminology, the reference frame is not stated explicitly for each component. The manuscript would profit from such explicit statements. Is the FACTS regional relative sea level rise N15 in Gregory et al. 2019? Are the components in the geocentric reference frame? Clarification on the reference frame will help scientists to add new modules.

We add a note: "Consistent with IPCC AR6, for existing sea-level component modules, the standard convention is that output is relative to the 19-year average of global-mean and/or relative sea level centered in the year 2005."

4) FACTS only works for the standard RCP/SSP scenarios except for workflow 1e if I understand it correctly. This is different to sea level emulators and I understood it only late in the text. This should be made more prominent so readers can better contextualise this work.

This depends on the modules being used. All the modules listed in Table 3 with a label other than "static by SSP scenario" or "static by RCP scenario" can be used with any emissions scenario. This includes all the modules used for global sea level projections in the medium confidence workflows described in the paper (wf1e, wf1f, wf2e, wf2f), though not the deconto21 or bamber19 modules. For RSL projections, this is generally also true, though the tlm/sterodynamic module does require the identification of a SSP climate scenario to use for calibrating the relationship between global-mean thermal expansion and ocean dynamic sea level. (This relationship can, however, be applied to temperature and ocean heat content output associated with any emissions scenario.)

Detailed comments (part 1):

l15-l18: can we include a non-US reference as well.

We now note: "Sea-level scenarios were also explored by the Dutch Ministerie van Verkeer en Waterstaat in 1986 (van der Kley, 1987)."

l29: introduce relative sea level rise and its definition (e.g Gregory et al. 2019)

Accepted, thank you.

l34: is relative sea level changes here the right term? So did Mitrovica already look into VLM influenced by West Antarctic ice loss or is it only about water mass redistribution?

Yes. Indeed, VLM influenced by West Antarctic ice loss was included in the original fingerprints published by Clark and Lingle 1977.

l38: this paragraph does not mention vertical land motion, a key component of local relative sea level rise.

We have replaced the reference to 'land subsidence' with a reference to 'vertical land motion.' Note that the 'D' of 'GRD' is solid-Earth deformation, i.e, a contributor to vertical land motion.

l47: "a single probability distribution"

Accepted, thank you.

l111: a word missing after MPI/OpenMP?

We have rephrased to say that tasks "can run on single or multiple cores, nodes, and threads."

l109-l142: Why this detailed description of RADICAL-Cybertools? It distracts from the story and does not help to get the code running. I would revise and shorten this and describe the environment in terminology understandable to sea level and climate scientists. See also part 2 of the review.

We have shortened this section somewhat, but this paper is a description describing a modeling framework, and we believe it is important to describe the architectural underpinnings of the framework.

l145/Figure1: what do the abbreviations like WF1e in the integration and extreme sea level step mean? They are not explained here.

Added a note to the caption that 'WF' is the acronym for Workflow. Caption already referred reader to Table 2 for the modules making up each Workflow.

l167: "bring this formerly offline simulation within FACTS" is not good to understand. Please reformulate for clarity.

We now note: "In the case of `ipccar6/ismipemuicesheets` and `ipccar6/gmipemuglaciers`, climate output generated offline using FAIR by the AR6 Working Group 1 Chapter 7 authors was run offline through the emulandice emulator of Edwards et al. (2021), the output of which was then transferred to the FACTS modules as static data. Similarly, in the case of `ipccar6/larmipAIS`, the Chapter 7 climate output was run through the LARMIP-2 emulator of Levermann et al. (2020), then transferred to the FACTS module. (Details of both the emulandice and LARMIP-2 emulators are described below.) For replicability reasons, the original AR6 direct-sample version of the emulandice ISMIP6 and LARMIP modules (`ipccar6/ismipemuicesheets` and `ipccar6/larmipAIS`, respectively) are retained in FACTS 1.0, though their use is deprecated…. In FACTS 1.0, the `larmip` and `emulandice` modules bring the formerly offline emulandice and LARMIP-2 emulators into FACTS."

l170: "demonstrate the ability"

Accepted, thank you.

l185: "an additional basal ice shelf melting" can this be more concrete with a number?

We now state: "16 state-of-the-art ice-sheet models performed experiments in which they applied a constant additional basal ice shelf melt forcing of 8 m/yr underneath each of five distinct regions of the Antarctic coast for 200 years."

l187: convoluted→convolved?

Accepted, thank you.

l187-192: I would reorder the sentences so the order represents the causal chain from global mean temperature change to ice loss. As of now a bit hard to follow.

We now state: "To apply these linear response functions to generate new projections, global mean temperature projections are scaled and time-delayed in according with the response of the CMIP6 climate models' subsurface oceanic warming to surface warming. This subsurface warming signal is then scaled with the observed sensitivities of basal melting to warming outside of the Antarctic ice shelf cavities. The resulting basal melt forcing is convolved with the linear response function to project the dynamic response of the Antarctic ice sheet."

l198ff: it is not clear to me from this paragraph if the authors implement a method already present in the AR5 or if they create a method in FACTS to capture the numbers of 2005-2010 observed and 2100 projected ice loss of the AR5.

We now clarify: "This is done within the `larmip` module using the same approach as applied by the IPCC Fifth Assessment Report (Church et al., 2013) and in the `ipccar5/icesheets` modules, described below."

l204: "a negative rate is added": can you say this more precisely?

We now state: "a negative rate term that scales with accumulation is added to account for the feedback between enhanced accumulation and dynamic ice discharge."

l219: appled→ applied

Accepted, thank you.

l219 "were applied in the context of the corresponding" is not clear. Do you mean "the RCP scenario projections were treated as SSP scenario projections"?

Yes, as noted, "e.g., RCP2.6 projections from DeConto et al. (2021) applied to SSP1-2.6."

L220: fingerprints precomputed, do they include the ocean bottom deformation part?

Yes, that is the 'D' part of 'GRD'.

l221: "includes"→"include"

Accepted, thank you.

l221ff: do the GRD fingerprints only influence the geocentric part of relative sea level rise or also VLM?

The 'D' in 'GRD' refers to the deformation of the solid Earth, i.e., land motion. We now clarify: "These fingerprints include both gravitational and rotational effects on sea-surface height, as well as deformational effects on sea-floor height."

l227: "of 2015-2100 glacier loss" or similar;

Accepted, thank you.

L227: which RCP scenarios are used?

We now state: "under RCP 2.6 and 8.5, and in some cases also under RCP 4.5 and 6.0".

l233: fl(t)^p: using f for the parameter here is confusing, it reads like a function, maybe write "f x I(t)" or choose another parameter name.

We have added a multiplication sign. We are sticking to the original symbology for consistency with the AR5 supplementary material.

l235: "a set of glacier models": can you be more specific?

We now state: " a set of four glacier models (Giesen and Oerlemans, 2013; Marzeion et al., 2012; Radić et al., 2014; Slangen and Van De Wal, 2011)."

l233ff: if readers do not cross this paragraph, they do not understand that the method used in the AR6 is named ipccar5, but uses an updated calibration.

We add a footnote to Table 1: "The `ipccar5/glaciers` module includes, in addition to the original IPCC Fifth Assessment Report calibration, recalibrations to GlacierMIP and GlacierMIP2 (Hock et al., 2019; Marzeion et al., 2020). The GlacierMIP2 recalibration is used in IPCC AR6 and in this paper and is denoted by a parenthetical '(GMIP2)' in this paper's tables."

l246ff: are these GRD fingerprints?

Yes, clarified.

l251: what does "tlm/" abbreviate?

'Tlm' abbreviates 'two-layer model', which describes the representation of temperature (from Geoffroy et al., 2013) used in the configuration of FaIR. We now note: "(As noted above, fair/temperature is run using a two-layer model representation of the forcing/temperature coupling, from whence comes the abbreviation 'tlm.')"

l259: where is the dedrifting and regridding documented to reproduce the work?

These are described in the supplemental material to IPCC AR6 WG1 ch. 9. Now noted.

l264: "is then projected"

Accepted, thank you.

l266: "projects global mean thermosteric sea-level rise, taking as input ... global mean thermosteric sea-level rise …" is confusing to read. I suggest to revise this sentence.

We clarify: "As described in Church et al. (2013)}, the ipccar5/thermalexpansion module projects the distribution of global mean thermosteric sea-level rise. It is calibrated to the time-dependent mean and standard deviation of the global mean thermosteric sea-level rise simulated by a multi-model ensemble."

l283ff: learning here that all earlier described components do not include VLM, so they do not output relative rise. It would be good to make this explicit before. It would be also good to say on which reference system all the other components work.

This is an unclear statement. The VLM module provides long-term VLM (and SSH change) from sources unrelated to contemporary land ice and land water redistribution. VLM associated with contemporary GRD effects are included in the GRD fingerprints. Now clarified. "Because the statistical model is constructed to extract a century-scale, climate-uncorrelated trend, there should be minimal double-counting of the deformational effects associated with recent land-ice mass loss and land-water redistribution. Vertical land motion associated with future land-ice mass loss and land-water redistribution is incorporated into the GRD projections of those components' respective modules."

L285: "constant trend": this means that future VLM is independent of future warming. Good to say this more explicitly.

Now clarified, per above.

l307: "Below the support" is hard to understand. Rephrase.

Rephrased as "Below the threshold of the Generalized Pareto Distribution."

l314: ", with the substitution …" this part of the sentence is hard to follow. Rephrase.

Now note: "As previously described in the description of the IPCC AR6 land-ice modules,  in FACTS 1.0, we substitute of the temperature-driven emulandice and larmip modules for the approach of direct-sampling offline calculated values used in AR6."

l323/Table1: module names ipccar5/ and ipccar6/ suggest that these are the ones used in the respective IPCC reports and the others not, but this is not the case following the text. This should be made clear in the caption.

We now note: "The `ipccar6` modules are direct-sample modules that were used only in IPCC AR6, and have been deprecated in FACTS 1.0 in favor of the `emulandice` and `larmip` modules. The `ipccar5` modules indicate the methods of described in (Church et al., 2013b), which in some cases and contexts were used by AR6, as described in (Fox-Kemper et al., 2021a) and Table 2. The `ipccar5/glaciers` module includes, in addition to the original IPCC Fifth Assessment Report calibration, recalibrations to GlacierMIP and GlacierMIP2 (Hock et al., 2019; Marzeion et al., 2020). The GlacierMIP2 recalibration is used in IPCC AR6 and in this paper and is denoted by a parenthetical '(GMIP2)' in Tables 2 and 3."

l327: can we give a more precise ref than just AR6? is it the unshaded cells in Table 9.9?. fullstop missing after the reference.

These are presented in numerous figures and numerous points in the chapter 9, Technical Summary, and SPM text. We have added "(for example, in the unshaded columns of Table 9.9)".

l329/Table2: can we mark here which are the workflows for IPCC AR6 projections?

All 7 workflows are used by AR6. The particular p-boxes into which they are incorporated are described by the second paragraph of 2.4. We have added sub-headers to the develop to make the distinction between the *medium confidence* and *low confidence* workflows.

l331: is it the shaded last row of Table 9.9 AR6 WG1? Please reference.

Low confidence projections for SSP5-8.5 are the shaded last column of Table 9.9. Low confidence projections for SSP1-2.6 are also calculated and shown in certain figures and

the text, but not presented in the table, as they are not substantially different from the *medium confidence* projections. We have added: "(for example, for SSP5-8.5, in the final column of Table 9.9)".

l344/45: this means this is not a full emulator as FACTS cannot map global mean temperature to sea level rise. Depending on modules it is restricted to RCP/SSP scenarios.

FACTS is a framework, not an emulator. Some modules are emulators. We have added some examples: "Some sea-level components modules (for example, the sterodynamic, ice sheet, and glacier modules used in workflows 1e, 1f, 2e, and 2f) take the FaIR-projected warming as an input. Others rely upon pre-computed projections, in some cases indexed by SSP or RCP emissions scenario (for example, the `deconto21` and `bamber19` ice sheet modules, and the deprecated `ipccar6` ice sheet and glacier modules) (Table 1)."

L346/Table3: the reader is left alone how these numbers compare to AR6. It would help to present the AR6 numbers again in Table 3/4 and discuss deviations in a paragraph.

We have added an appendix table showing the component-wise comparison to AR6. As can be seen, the FACTS 1.0 modules agree with results presented in AR6 within 0.01 m rounding errors. Discrepancies in total results can be slightly larger (on the order of 2-3 cm), consistent with a combination of rounding errors and sampling differences. (Note that AR6 used 20,000 samples per workflow, compared to the 2,000 per workflow in the results shown here.)

l357: cm→m

Accepted, thank you. Converted all sea level measurements to m for consistency.

l358: what are "Workflow pairs"?

We now clarify: "other emulandice/parametric Workflow pairs (i.e., 2e vs. 2f, and 3e vs. 3f)"

L372ff: please add a reference or an explanation of how the projection variance and interaction terms are calculated.

We have rewritten the caption for Figure 4: "Variance decomposition of GMSL change in 2100 under SSP1-2.6 (left column) and SSP5-8.5 (right column), under Workflows 1f, 2f, 3f and 4, in the style of Hawkins and Sutton (2009). Each colored wedge in this figure represents the variance across Monte Carlo samples for a particular component, under the specified scenario and Workflow, normalized by the variance of projections for total sea-level change in the same scenario and Workflow. The difference between the sum of component variances and the total variance (normalized to 1.0) represents the interaction among components." Parallel changes were made to the caption for Figure 7.

l405: explain TE

Thermal expansion. Added acronyms used in the figures to the caption.

L410: indeed I would see it as a major aim of the manuscript to replicate the main AR6 SLR projections.

This is not the primary aim of the authors. The aim of the manuscript is to describe an updated version of the modeling framework used in AR6. The exact scripts and data sets used for the AR6 analysis, while harder to use than FACTS 1.0, are available at https://zenodo.org/record/6419954. However, the newly added appendix tables show that FACTS 1.0 replicates the AR6 results, with minor differences in rounding and sampling.

L417: why the difference in how likely ranges are defined in this study compared to the rest of the IPCC AR6? Can the motivation be stated?.

This is beyond the scope of this paper, which is a model documentation paper. It is addressed in Kopp et al. (2023), which describes the sea-level uncertainty and ambiguity framing adopted by AR6. In brief, due to the deep uncertainty in sea-level projections, it is not possible to define precise probability ranges for outcomes, and the imprecise probabilities associated with the canonical IPCC definitions of likelihood are more appropriate.

L450: with some caveats: can you detail how this was translated?

AR6 states: that it is "mapping 2°C and 5°C stabilization scenarios to SSP1-2.6 and SSP5-8.5, respectively." Referring to 2300 projections, it states that "Incorporating the SEJ-based ice-sheet projections of Bamber et al. (2019) for 2°C and 5°C stabilization scenarios yields 1.0–3.1 m for SSP1-2.6, and 2.4–6.3 m for SSP5-8.5, although because of the differences in scenarios, the SSP1-2.6 estimates may be overestimated and the SSP5-8.5 may be underestimated." The same is already noted in the parenthetical in the text: "(though SSP1-2.6 most likely stabilizes below 2°C and SSP5-8.5 continues after 2100 to warm well above

5°C)".

L474: "... for glaciers". I am not sure if this is generally true. Also glaciers have different timings of mass loss and disappearance in different world areas.

Since there are separate fingerprints for the RGI glacier regions, different timings of mass loss and disappearance do not invalidate the point. We have clarified: "Such a library approach is most appropriate for glaciers, as the glacier regions are geographically small enough that the shifts in the locus of mass loss within a region will not substantially modify that region's fingerprint."

Part 2:

The authors state in I71 that "FACTS 1.0 allows replication of the AR6 approach entirely within FACTS", but I did not manage to make the code work on our computers. A main hurdle is the EnTK framework, which seems to be a specific framework only installed on certain supercomputers. Making this the default option to run FACTS hinders most scientists from replicating the work. I would therefore advise to change the default option for running FACTS to something generic many scientists are accustomed to. The authors provide a blueprint for such an option via a shell-script. I recommend making this the default option, or provide an alternative approach that can be used to reproduce AR6 numbers. In any case, the code should be cleared of hardcoded paths (e.g. using one configuration file shared across modules, or one per module with a consistent format across modules) and the authors should better describe how R should be installed to make the land ice emulators work. Also provide a description how to reproduce the AR6 numbers within the code/Readme. Ideally FACTS would in addition be provided as a package and could be installed using the usual tools (pip install or similar).

Concerning the manuscript, a clear reference to the AR6 numbers facts aims to replicate is missing. I expect these are Table 9.9 AR6 WG1. One solution would be to add them to Table 3 and 4 of the manuscript for direct comparison. I also recommend to provide computational cost per module (CPUh or similar) so potential users of the framework can judge if installation of the EnTK framework is necessary.

Detailed comments (part 2):

The code base includes a large number of dependencies, including heavy dependence on R. The authors provide some guidance in how to install the dependency, but it appears set up for their specific system, and not particularly user-friendly for the larger scientific community. In particular, library location and work directory are hard-coded in files disseminated throughout the project (in individual modules), as opposed to clearly indicated in a centralised configuration file.

For instance, to run the config provided in the doc:

cp -r experiments/coupling.ssp585/config.yml test

python3 runFACTS.py test

first fails because of missing files in modules/emulandice/shared (emulandice_1.1.0.tar.gz and emulandice_bundled_dependencies.tgz). To produce them, it was necessary to set up a local R environment. This required among other things to edit:

modules/emulandice/shared/emulandice_environment.sh : the line "module use /projects/community/modulefiles" had to be commented out.

modules/emulandice/shared/emulandice_bundle_dependencies.R:

packrat::set_opts(local.repos = c("/projects/community/R3.6_lib_workshop","."))

packrat::install_local('cli')

…

Needed to be replaced with more traditional:

install.packages('cli')

The authors did provide a README file in that directory with the mention:

"You will likely need to customize emulandice_environment.sh and emulandice_bundle_dependencies.R based on your local environment."

But we recommend the default to be setup for generic linux system, and "customization" reserved for use on the authors HPC, instead of (currently) the opposite.

This is a particular issue for the *emulandice* module, which was written in its current form in part to demonstrate how a FACTS module can wrap around independently developed code (in this case, the *emulandice* code of Edwards et al., 2021, https://github.com/tamsinedwards/emulandice/), even if written in a language other than Python. Of course, this does create some challenges, because the system needs to be set up to run this independently developed code.

A future version of FACTS will include a module package management system to facilitate this task. For the current version, we will improve the documentation for setting up *emulandice*.

To our knowledge, no other module set has similar issues.

To improve ease of use, we have (1) adopted *emulandice* defaults that work for a local Linux install, (2) created a single *emulandice_build.sh* script, and (3) added instructions for configuring this particular module set to the Quick Start documentation.

We are also creating a Docker container for FACTS, and adding instructions for running FACTS within a container to the documentation.

> The EnTK framework is the largest hurdle for use in the wider scientific community. The welcomed, alternative –shellscript option is experimental. We identify it as the main area to improve in order to disseminate the work.

> runFACTS.py (issue with the EnTK framework)

> The Mongo DB Server installation was smooth following the instructions provided by the authors. However, we quickly ran into issues with their EnTK framework when following the documentation:

> python3 runFACTS.py experiments/dummy

> "radical.entk.exceptions.EnTKError: Shell on target host failed: Cannot use new prompt,parsing failed"

> The authors offer an alternative (https://fact-sealevel.readthedocs.io/en/latest/quickstart.html#testing-a-module-with-a-shell-script) where the code produces a shell script to bypass the EnTK framework, but with a strong disclaimer ("Performance is not guaranteed, and multi-module experiments are very likely not to work without customization.").

> I tested the dummy setup and had to make minor modifications to runFACTS.py:

> - print(' WORKDIR=/scratch/`whoami`/test.`date +%s`')

> + print(' WORKDIR=local_scratch/`whoami`/test.`date +%s`')

> - print(' OUTPUTDIR=/scratch/`whoami`/test.`date +%s`/output')

> + print(' OUTPUTDIR=local_scratch/`whoami`/test.`date +%s`/output')

> And create the local_scratch folder.

Then:

python3 runFACTS.py --shellscript experiments/dummy > test_dummy.sh

source test_dummy.sh

ran without error, but also produced no output.

I then tried the other configuration file indicated in the documentation, again with the –shellscript option:

mkdir test

cp -r experiments/coupling.ssp585/config.yml test

python3 runFACTS.py --shellscript > test_coupling.sh

source test_coupling.sh

I ran into issues related to library installation and hard-coded paths as described in the previous section. Once overcome, the script ran but new error messages appeared:

> cp: cannot create regular file 'local_scratch/reviewer/test.1681896393/output': No such file or directory

> cp: target 'local_scratch/reviewer/test.1681896393/test.GrIS1f.FittedISMIP.GrIS' is not a directory

(local_scratch is a local folder I created to replace the hard-coded, and authors-specific architecture /scratch)

Given the disclaimer provided by the authors on the experimental nature of the – shellscript option, I did not attempt to to run this further.

The –shellscript option is developed to facilitate testing of modules by module developers. EnTK handles process and file management for FACTS, a deliberate decision to facilitate the computational scalability of the code. We are unsurprised that the attempt to use shellscript mode for a coupled run failed, as nothing in the shell script takes on the file management role fulfilled by EnTK. Developing an alternative coupling framework to EnTK is not in the development pathway for FACTS.

EnTK is primarily developed to support executions at large scale on high performance computing (HPC) platforms and requires executing within a dedicated Python virtual

environment on a GNU/Linux operating system. For small runs and testing purposes, EnTK can run within a virtual machine or container on diverse operating systems. We regret the reviewer had challenges using EnTK.  We were unable to replicate their challenges; the authors were able to install EnTK and FACTS from scratch in a vanilla Ubuntu Focal Docker container on a Mac laptop, as well as on a Windows laptop with WSL2, and get it running with no similar issues. See *scripts*/*vm_factsenvsetup.sh* for an example of how to do this.

We are also creating a Docker container to ease the installation of FACTS for individuals who would like to run in a container.

Unfortunately, given the limitations of the anonymous peer review process, it is challenging to help the reviewer diagnose why an install that works on a vanilla virtual machine does not work on their system, but we would encourage them to try it out within a Docker container or VM if it does not work on their system. Further, we extended FACTS' documentation (https://fact-sealevel.readthedocs.io/en/latest/quickstart.html), adding (1) details about installation on a GNU/Linux workstation, virtual machine and container; (2) links to the relevant RADICAL tools documentation; (3) overall reviewed instructions.

**Referee #2: Luke Jackson [June 15 2023]**

**General Comments**

This paper outlines a modular platform designed to harmonise and internally calculate (tidal-datum-epoch) mean sea-level contributions from all major global and local sea-level components that are dependent upon a climate model emulator, and post-processed to localise and applied to extreme water-levels. The modular structure enables different combinations of sea-level component emulators/datasets within a fully probabilistic framework to be enabled thus accounting for aleatory and epistemic uncertainty. The GSL and New York example demonstrate the utility of the framework effectively. Scientifically, the framework benefits from more than a decade of research that exploits the budgetary approach to probabilistic sea-level change developed by numerous researchers. The framework also shows great potential and flexibility – I hope this will become an evolving resource that the community can utilise in future.

Overall, this is a welcome piece of work to the research community. It is carefully written and, in most places, clear to follow. There are a number of places where additional detail is required, ideally an additional case study would be shown, and a few Figures/Tables need updating.

**Specific Comments**

The Introduction has a strong IPCC focus. While this provides context, additional references to key work is important – particularly the development of a budgetary approach to SL, which is essential to the process-based method of projection.

Per the reviewer's suggestion below, we have added a reference to Slangen et al. (2012) in the narrative leading up to AR5. We have also clarified that the reference to Fox-Kemper et al. (2021) describing the "numerous subsequent studies" is to the overview in section 9.6.3.1. We have also added: "Examples of open-source probabilistic sea-level projection frameworks include the ProjectSL/LocalizeSL framework (Kopp and Rasmussen, 2021) developed by Kopp et al. (2014, 2017), and BRICK (Wong et al., 2017). Additional studies present probabilistic RSL projection methodologies without associated open-source software releases (e.g., Slangen et al., 2014; Grinsted et al., 2015; Jackson and Jevrejeva, 2016; Le Cozannet et al., 2019; Palmer et al., 2020)."

The choice of Workflows presented focuses specifically on combinations of AIS/GrIS emulations/outputs. The results certainly clarify the IPCC decision-making process (medium/low confidence) but showing a more diverse selection of modules would showcase the FACTs framework more effectively as an **assessment tool** (e.g., VLM). Adding a separate city-level case study where VLM is focused upon (in

addition to the current NYC example showcasing ice sheet combinations) would be valuable.

We appreciate the reviewer's suggestion, but the emphasis in this manuscript is on the distinctive approach FACTS takes to allow exploration of structural uncertainty. It provides an accurate representation of the current state of the code, which was strongly influenced by the needs of AR6. It is possible for users building off of FACTS develop additional modules, but at the moment we have only one VLM module (kopp2014/verticallandmotion) with global coverage. We have clarified that the alternative VLM module (NZInsarGPS/verticallandmotion) simply directly samples gridded land motion data from an external file. We note that, in Naish et al. (2022), this module applies a gridded data file describing rates of land motion inferred from interferometric synthetic aperture radar (InSAR) data. We do not add an additional case study, but some of the challenges in thinking about appropriate VLM projections to incorporate into RSL projections are described in Naish et al. (2022).

> The issue of a common reference timescale is only partially addressed. This is pertinent given your mention of IPCC AR5 (ref timescale 1986-2005) and some sea-level components (e.g., GrIS SMB Fettweis et al. 2013) that are relative to an alternative baseline (e.g., ~1970s). Highlighting this earlier and how this is dealt with (either as a post-processing step or module specific step) is very important. Likewise, you need to explain how/if this timescale can be user defined.

This is a convention that must be consistent across modules. It is standard for modules to have 'baseyear' as a parameter (for AR6 examples, this is set to 2005, the midpoint of the 1995-2014 reference period).

We now note: "Configuration options such as the number of samples to run, the time points at which calculations are reported, and the reference period used for output can be globally specified but are implemented on a module-by-module basis." In the discussion of directions for improvement, we state "The existing FACTS modules start projections in the 21st century." This is less specific than previously, as – while 2005 (1995-2014) is used as the center point for the results presented here – most of the modules can handle a centerpoint at any time after the year 2000 without problems. The point of the discussion is the absence of historical data, not the way in which reference periods are handled.

> Vertical Land Motion needs additional detail, and the issue of double counting needs to be addressed in the framework. If the barystatic fingerprints to be scaled use RSL, then they will contain a VLM component (as does the GIA RSL fingerprint). Scaling and summation of SL components within the third "Step" of the framework, would then induce possible double counting if the VLM component has not been corrected for this (arguable small) component. This would be true of K14 VLM or NZ VLM, as you allude to in the discussion.

This is addressed in the response to reviewer 1. We have clarified that we use the term 'vertical land motion' in the previous draft to refer to long-term vertical land motion, of the sort reflected in century-scale analysis of tide-gauge data. Elastic deformation associated with contemporary land-ice and land-water mass redistribution is accounted for in the RSL projections via the static GRD fingerprints. For the K14 module, double-counting is an issue only to the extent that there are substantial 20th century trends in deformation associated with century-scale barystatic effects. Given the magnitude of 20th century average barystatic trends (order 1.0 mm/yr), this effect will introduce a bias in most places of <±0.2 mm/yr, or 2 cm/century. This is quite small relative to other sources of uncertainty, and comparable to sampling issues.

Double counting with respect to GIA Is not an issue for the K14 method; this is taken into account in the methodology.

**Technical Corrections**

Line 5: rephrase "a modular … sea-level rise", needs to allude to individual SL components to generate global, regional, ESL projections.

We now state " the drivers of sea-level change and their consequences for global mean, regional, and extreme sea-level change."

Line 35-37: Highlight earlier work of Slangen et al. (2012) that led into AR5.

Added.

Line 43: "They also … core elements" – rephrase it is not clear what you mean here.

Rephrased "core elements "as "sea-level drivers".

Line 44: "these studies" – which studies? No reference to specifics or examples.

Rephrased as "in probabilistic sea-level projections."

Line 44: K14 and K17 do not refer specifically to "ProjectSL/LocalizeSL Framework" despite these being associated with the coded release. Rephrase to be more general and highlight additional frameworks beyond K14/K17.

We have added a reference to the LocalizeSL code archive. As this is a model development paper, we believe it is important to distinguish between software frameworks and published methodologies without associated software. We have rephrased: "Examples of open-source probabilistic sea-level projection frameworks include the ProjectSL/LocalizeSL framework (Kopp and Rasmussen, 2021) developed by Kopp et al. (2014, 2017), and BRICK (Wong et

al., 2017). Additional studies present probabilistic RSL projection methodologies without associated open-source software releases (e.g., Slangen et al., 2014; Grinsted et al., 2015; Jackson and Jevrejeva, 2016; Jevrejeva et al., 2019; Le Cozannet et al., 2019; Palmer et al., 2020)."

> Line 47: "This assumption …" rephrase for grammar, and an example (ideally non-SL to aid the reader) would be useful.

We have rephrased: "By definition, this assumption is not true for processes characterized by ambiguity."

> Line 49: remove "in", and add "different components" to "different sea-level components"

Accepted.

> Line 69: Slangen et al. in press – now published – update

Updated.

> Line 82: Terminology "Steps" versus "step" – is there a different between these?

We have updated to use "Experiment Step" consistently throughout.

> Line 95: "total" – you can only refer to total if all SL components at an appropriate scale are accounted for.

Rephrased as "their combined contribution to sea-level change".

> Line 103: "combined" – the notion of combining workflows needs articulating – this is not explained in the examples either – do you mean something like Sweet et al. (2022)?

Added "(for example, in a p-box, as discussed in section 4.1)")

> Line 113: "RADICAL-SAGA" is not discussed in the manuscript – what is it's purpose?

Streamlined this discussion per reviewer 1. Discussion of RADICAL-SAGA is deleted.

> Line 124: "exposes" – what do you mean by this?

This is standard terminology for allowing others to access features provided by a code via an API.

> Line 124/5: "Those constructs … tasks" is effectively repeated on line 128-129 – consider merging for clarity

This section has been streamlined.

> Line 125: Terminology "Ensemble" appears here then isn't mentioned elsewhere.

Dropped the term.

> Line 131: "failures of Tasks …" – what about failed Stages or Pipelines – how does the framework deal with these higher-level issues?

The description of EnTK has been streamlined, so this has been removed. Failures occur at a Task level, but if output needed by another part of a Stage is not produced, subsequent Tasks will also fail.

> Line 137-140: RADICAL Tools, their roles, Pipeline, Stage and Task need to be included within the schematic diagram (Figure 1) to better communicate section 2.2

Per Reviewer 1, this section has been streamlined. We decline to revise Figure 1, since it better conveys the information needed by a user in its current form, without diving into the development details of EnTK.

> Figure 1: see comment above. Including the 7 workflows discussed is of limited value if this is a conceptual diagram – a number of "example" workflows showcasing Task, stage and Pipelines would serve better.

We have clarified in the caption that this is a schematic illustration of the FACTS Experiment described in this manuscript.

> Line 148: "emissions scenario as input" – this is the only input/variability that is feasible within this module? For example, how many ocean layers are used (is it a default FAIR setup?)?

Clarified that this is using the AR6 calibrated and constrained parameter set. It would of course be possible to modify the module to allow other inputs, but as constructed emissions scenario is intended to be the only input. Clarified that this is using the two-layer temperature function of Geoffroy et al. (2013).

> Section 2.3.2 would benefit from each sea-level component being within a separate subsection (e.g., Section 2.3.2.1 Generic) rather than an italicised header that is inline with the main text.

Accepted.

Line 157: facts/directsample : does not appear in Figure 1

Several modules do not appear in Figure 1, which illustrates the FACTS Experiment used to generate the output in the paper, analogous to that in AR6.

Line 165: FAIR 1.0 – should this be FACTS1.0 ?

Corrected, thanks.

Line 168/169: "The emulandice …" new paragraph

Accepted.

Line 170: "simulations and demonstrates" to "simulations. These demonstrate"

Modified.

Line 184: does Levermann et al. 2020 refer to the original code or the modification to speed it up? If the former then move the citation prior to the comma in Line 183.

Accepted.

Line 196: "fixed rate of mass loss" – does this rate refer to basal melting or SMB or both?

Ice-sheet mass loss. Now clarified.

Line 196/197: "This assumption, … underestimate …" Does it? By how much?

Thank you for flagging this issue. While in AR6 we made the conservative assumption that no acceleration happened in the LARMIP-driven runs beyond 2100, in fact it is possible to adjust the module to allow for future accelerations/decelerations, and we have done so. We've therefore removed this text from the paper, and instead state: "Whereas in AR6, LARMIP-2 projections (including surface mass balance) are extrapolated beyond 2100 assuming a fixed rate of ice-sheet mass loss after 2100, here we allow the rate of loss to evolve following the linear response function formulation."

Line 214: "through" to "to"

Respectfully disagree.

Line 215/216: "In IPCC AR6 … projections" What do you mean by "applied in the context of" – this is ambiguous.

Changed to "mapped to"

Line 219: "were appled" (spelling) and rephrase – sentence reads oddly – perhaps alternative wording to "applied"

Changed to "employed".

Line 221: "includes" to "include"

Accepted.

Line 221: "regional scaling" to "regionalisation of their global SL equivalent contribution" (or similar)

As this is a scaling (i.e., the fingerprints multiply the global projection), we leave as is.

Line 224: needs more information on the fingerprints (e.g., all barystatic components – grouped or separate or both such as AIS or WAIS, APIS, EAIS or by-sector, realistic (observed) or uniform, range of earth properties or not?)

All the existing ice-sheet modules include in their post-processing stage a regional scaling based on GRD fingerprints for West Antarctica, East Antarctica, and Greenland (e.g., Mitrovica et al., 2001; Gomez et al., 2010; Mitrovica et al., 2011). The fingerprints include both gravitational and rotational effects on sea-surface height, as well as deformational effects on sea-floor height. They are implemented as static fingerprints that do not change over time; as in Kopp et al. (2014), mass change is assumed to be uniform across the respective regions. The fingerprints were pre-computed (outside the FACTS framework) by solving the sea-level equation with a pseudo-spectral approach up to spherical harmonic degree and order 512 (equivalent to a spatial resolution of about 0.4°). They assume a radially symmetric, elastic and compressible Earth model, based on the Preliminary Reference Earth Model (Dziewonski and Anderson, 1981).

Line 239/240: Remove "(In this manuscript, …)"

We believe this parenthetical comment is an important note for the reader to be able to reproduce the results in the paper.

Line 257: "(Fox-Kemper et al., 2021))" to "Fox-Kemper et al., (2021)"

Accepted.

Line 276: "The model-based …" – not clear if this relates to module or K14 method

We clarify that "The groundwater projection of Pokhrel et al. (2014), based upon a water resource assessment model, is included as an option for sensitivity analysis."

Line 288/289: "Sensitivity tests …" – are these published somewhere? If not, then inclusion within an Appendix would be very useful.

We have added an appendix figure showing the difference if the Kopp et al. (2014) methodology if applied using a prior based on ICE-6G-C VM5a instead of the ICE-5G VM2-90 used in the standard methodology.

Line 290: "The spatiotemporal model" – new paragraph but I assume you are talking about VLM? Needs restating.

Adjusted paragraph spacing.

Line 293: "tuned" – what do you mean by this? Is this a scaling procedure/process to minimise misfit to observations?

This is a maximum likelihood optimization. Now clarified.

Line 295-297: much more detail on NZInsar module needed as vertical reference frame of InSAR/GPS differs from RSL benchmarks regionally. The way rates etc from this module are harmonised is important to explain. Likewise, direct VLM observation will contain components of VLM from each barystatic SL component and GIA – how are these accounted for (prior or post Step 3)?

Details of the InSAR analysis (described in Naish et al., 2022) are beyond scope for this manuscript; important for this manuscript is that the module imports a file describing an independently derived field of VLM rate estimate.

Line 306: "Annual means … stage" place this before fitting sentence to order processing steps correctly.

Accepted.

Line 308: "Below the … (Buchanan et al., 2016)." I didn't understand this sentence – sorry – rephrase

Rephrased: "Below the threshold of the Generalized Pareto Distribution, a Gumbel distribution with support between Mean Higher High Water and the threshold is assumed and used to compute return periods."

Line 311: Need a statement here that the dynamic evolution of ESL (due to changing atmos-ocean clim. Variability) is not accounted for in the current framework.

Accepted.

Line 313: Remind the reader of Workflows here.

Added "(i.e., sets of sea-level component modules)".

> Line 333: "pseudo-random" – what do you mean by this?

"Pseudo-random" is a term used to describe nominally 'random' numbers generated by a deterministic process (e.g., by the python module `random`).

> Line 338: temperature should include a ° symbol

Accepted.

> Line 358: "cm" to "m"

Accepted.

> Line 370: stay consistent with units – switch to m

Accepted.

> Table 3: add GSAT into title; rewrite footer for consistency so that T and GMSL are clearly identified with different baselines rather than using "except"; Use an asterisk to refer to specific modules that use complimentary RCP rather than the SSP, and refer to this in the footer – reader doesn't have to go hunting elsewhere in the text.

Done.

> Figure 3: Present time to 2100 rather than 2150 so that it is consistent with other Figures. Since you are not showing the uncertainty change shape after 2100 the thick/thin bars could mislead the reader; caption spelling mistake "think" to "thin".

Accepted.

> Line 457-464: Missing detail on GIA (as distinct from VLM), such as GIA uncertainty (e.g., Melini & Spada, 2019) and present-day GRD uncertainty (e.g., Cederberg et al. 2023).

GRD uncertainty is addressed later in the discussion; we have added a reference to Cederberg et al. (2023).

While GIA uncertainty is significant in explaining 20th century sea-level budgets, and critical to explaining pre-industrial RSL proxies, Melini & Spada (2019) show uncertainties of < 0.5 mm/yr in most of the world, which is small in the context of sea-level projections. Moreover, the VLM approach of Kopp et al. (2014), as we now make clear, also picks up long-term geocentric sea-level changes from GIA; thus, GIA uncertainty should be significant in the current projections only in parts of the world where (1) it is significant in the models (mostly

polar regions), (2) it cannot be constrained by 20th century observations, and (3) it is large in comparison to both other drivers of VLM and contemporary GRD.

We now note: "The current VLM modules also do not explicitly address uncertainty in GIA (e.g., Melini & Spada, 2019). In the kopp14/verticallandmotion module, GIA uncertainty does not making a substantial contribution in locations where tide-gauges are available to constrain long-term changes; however, this uncertainty can be significant at sites distant from tide-gauges, particularly in polar regions where the GIA contribution is largest (Figure A1). New approaches to fusing model projections with geological, tide-gauge, and satellite observations could better characterize this uncertainty (e.g., Caron et al., 2018).

Line 462: "scenarios" – should mention non-linear VLM behaviour too.

We note that "A module capable of representing alternative scenarios of such factors and their evolution over time could be helpful in assessments in such regions."

Line 478-484: See Specific Comment on common timescales – emphasis on compatibility of component-based timescale including use of observation/projection-data to supplement historical runs to bridge the gap between modelling intercomparison projects remains important.

See response above.

**Changelog from FACTS 1.0-rc to FACTS 1.0**

- updated emulandice scripts for a more generic system
- rename module functions for greater consistency
- improve install process, including set up that works in a vanilla Ubuntu VM
- added support to pickle larger objects and data types to sterodynamics modules to prevent overflow errors
- assured random shuffling of temperature samples in larmip projection stage
- fixed temperature sample correlation issues in sterodynamics and FittedISMIP modules
- more efficient use of xarray in totaling scripts
- fixed depreciation issues in emulandice
- **Created Docker container**
- **Added notebooks for generating p-boxes**

---

## Author Response (AR2)

11 November 2023

Dear Dr. Wickert,

Thank you for your work on the manuscript. We are very pleased with how the review process has improved both the clarity of the manuscript and the usability of the FACTS code base.

Below, please find our responses to your line-by-line comments. In general, we have accepted them, though there are a few cases regarding sea-level terminology and associated style choices where we have not. Instead, we have chosen to follow current best practice in the community studying modern sea-level change, as set forth in Gregory et al. (2019, 10.1007/s10712-019-09525-z). We have also opted for consistency with the terminology and acronyms employed the IPCC Sixth Assessment Report. We recognize, given the authorship of this manuscript, that this is a bit of justification by self-citation, but as Fox-Kemper et al. (2021) represents the most broadly cited recent global sea-level assessment and has undergone an extremely extensive expert and government review process, we believe consistency with it is justified. We have also added a text box to make key definitions more accessible.

> 7. second "alternative" --> different?

Accepted.

> 8. and thereafter: hyphen between sea-level when modifying "change". Hyphens should be used when multiple nouns are chained together to present a single concept.

We have hyphenated all occurrences of 'sea-level' as a compound adjective.

> 10. sheetS

Accepted.

> 15. 1982), and the first

Modified.

> 18: This sentence about the Dutch ministry seems a bit odd in standing alone, and is written in the passive voice without much context. Could this be better brought into the overall theme of the paragraph and prior (US-based) efforts?

Now rephrased: "The US Army Corps of Engineers and the Dutch Ministerie van Verkeer en Waterstaat first employed planning-oriented sea-level scenarios just four years later, in 1986 (US Army Corps of Engineers, 1986; National Research Council, 1987; van der Kley, 1987)."

> 39-40 and thereafter. Members of the scientific community studying GRD uses "GIA" to or "the GIA process" to describe GRD. I personally prefer GRD because it is direct and descriptive. That said, it might be good to explain with some precision how GIA is defined here.

We have added a box defining key terms based on Gregory et al. (2019).

41. Could you write out what "this" challenge is? Perhaps some infomration has fallen off of the end of the line in the PDF.

Clarified: "the challenge of producing comprehensive, localized RSL projections."

46. Is it worthwhile considering changes in groundwater storage as well, which may or may not relate to anthropogenic factors?

Potentially. In general, the literature has focused on anthropogenic groundwater storage changes because on modern timescales, trends in this term are likely to substantially outweigh any natural trends. However, some Earth system models do include a degree of representation of groundwater changes in response to climate forcing and variability, which are significant in terms of interannual variability.

61. "ESL" has historically been used for "Eustatic Sea Level". While that has fallen a bit out of favor, I wonder if "ExSL" might be less ambiguous.

In general, the literature uses "e.s.l." (lowercase) for 'equivalent sea level' or 'eustatic sea level', or "s.l.e." for 'sea level equivalent', in order to present changes in ice mass in global mean sea-level equivalent terms. AR6 uses 'SLE' for sea-level equivalent and 'ESL' for extreme sea level. The latter is very common in the modern sea-level literature. (281 citations in Google Scholar appear in response to the search for 'esl "extreme sea level"'.) We retain "ESL" for consistency with the existing literature.

The term 'eustatic sea level' is deprecated due to ambiguity; as Gregory et al. (2019) note, "*In recent literature, ''eustatic'' is often used as a synonym for ''barystatic'', whereas in geological literature eustatic sea-level change means either global-mean sea-level rise or global-mean geocentric sea-level rise. Because of this confusion of meaning, we deprecate the term ''eustatic'', following the last three assessment reports of the Intergovernmental Panel on Climate Change (Church et al. 2001; Meehl et al. 2007; Church et al. 2013).*"

119. What does "extreme" actually mean in "extreme scale"?

Modified because it is not especially germane to the present application. Now reads: "RADICAL-Cybertools are software systems designed to support the execution, across computing scales, of applications comprised of multiple tasks."

148. is --> contains

Accepted.

169. comma to avoid need for double-parentheses?

Accepted.

175-176. This stentence reads awkwardly to me. Maybe "Existing sea-level component modules output water-surface elevation as an anomaly from the 19-year average centered on the year 2005, for consistency with the IPCC AR6."

We state: "Consistent with IPCC AR6, existing sea-level component modules output quantities that are relative to the 19-year average of GMSL and/or RSL centered in the year 2005." 'Water-surface elevation' would not be correct, as the outputs are either the contribution to GMSL change or to RSL change.

197. offline --> offline-coupled ?

Accepted.

216 and others. provided in --> provided by ?

Accepted.

269. no comma before "based", I think

Accepted.

305. thermal expansion (add "thermal"), for clarity?

Accepted.

345. This is one place in which I am unsure about the GIA vs. GRD distinction.

Now Addressed in Box 1.

**Contemporary GRD:** GRD due to ongoing changes in the mass of water stored on land as ice sheets, glaciers and land water storage.

**Glacial isostatic adjustment (GIA):** GRD due to ongoing changes in the solid Earth caused by past changes in land ice.

Table 4: Global/Relative sea level CHANGE [in headers] ? Perhaps also provide units in the headers?

Accepted.

Thank you again for your work on the manuscript,

Bob Kopp

[revised manuscript text omitted]